# Towards agricultural soil carbon monitoring, reporting and verification through Field Observatory Network (FiON)

Olli Nevalainen[1], Olli Niemitalo[2], Istem Fer[1], Antti Juntunen[2], Tuomas Mattila[3], Olli Koskela[2], Joni Kukkamäki[2], Layla Höckerstedt[1], Laura Mäkelä[4], Pieta Jarva[4], Laura Heimsch[1], Henriikka Vekuri[1], Liisa Kulmala[1,5], Åsa Stam[1], Otto Kuusela[1,6,7], Stephanie Gerin[1], Toni Viskari[1], Julius Vira[1], Jari Hyväluoma[2], Juha-Pekka Tuovinen[1], Annalea Lohila[1,6], Tuomas Laurila[1], Jussi Heinonsalo[5], Tuula Aalto[1], Iivari Kunttu[2], Jari Liski[1]

[1] Finnish Meteorological Institute, FMI, Helsinki, Finland
[2] Häme University of Applied Sciences, HAMK, Hämeenlinna, Finland
[3] Finnish Environment Institute, SYKE , Helsinki, Finland
[4] Baltic Sea Action Group, BSAG, Espoo, Finland
[5] University of Helsinki, Institute for atmospheric and Earth system research (INAR), forest sciences, Helsinki, Finland
[6] University of Helsinki, Institute for atmospheric and Earth system research (INAR), physics, Helsinki, Finland
[7] University of Amsterdam, Computational Science, Amsterdam, Netherlands

*Correspondence to*: (olli.nevalainen@fmi.fi)

**Abstract.** Better monitoring, reporting and verification (MRV) of the amount, additionality and persistence of the sequestered soil carbon is needed to understand the best carbon farming practices for different soils and climate conditions, as well as their actual climate benefits or cost-efficiency in mitigating greenhouse gas emissions. This paper presents our Field Observatory Network (FiON) of researchers, farmers, companies and other stakeholders developing carbon farming practices. FiON has established a unified methodology towards monitoring and forecasting agricultural carbon sequestration by combining offline and near real-time field measurements, weather data, satellite imagery, modeling and computing networks. FiON's first phase consists of two intensive research sites and 20 voluntary pilot farms testing carbon farming practices in Finland. To disseminate the data, FiON built a web-based dashboard called Field Observatory (v1.0, fieldobservatory.org). Field Observatory is designed as an online service for near real-time model-data synthesis, forecasting and decision support for the farmers who are able to monitor the effects of carbon farming practices. The most advanced features of the Field Observatory are visible on the Qvidja site which acts as a prototype for the most recent implementations. Overall, FiON aims to create new knowledge on agricultural soil carbon sequestration and effects of carbon farming practices, and provide an MRV tool for decision-support.

# 1 Introduction

Farmers are managing one of the largest carbon stocks on the planet where even relatively small additions are important for climate change mitigation. Accordingly, the international "soil carbon 4 per mille" initiative aims at raising the soil organic carbon content by 0.4 % per year by adopting carbon farming practices (Minasny et al. 2017). Carbon farming practices include methods, such as increasing carbon inputs (soil amendments, cover crops, residue management) and crop rotations. Such practices do not only have the potential to partially refill the global soil carbon stock that has lost 116 Pg carbon due to land cultivation (Sanderman et al., 2017), but they could also improve soil structure and health, and increase crop yields (Merante et al. 2017; Oldfield et al. 2018). Annual carbon sequestration rates for different management practices vary from 100 to 1000 kg C ha$^{-1}$ (Merante et al., 2017; Minasny et al., 2017). Detecting sequestration rates in this range is difficult with traditional empirical soil sampling designs due to large spatial variability of soil carbon content and small relative changes in the soil carbon stock due to individual management actions (VandenBygaart and Angers 2006; Heikkinen et al. 2021). This calls for better monitoring, reporting and verification (MRV) of the amount, additionality and persistence of the sequestered soil carbon due to carbon farming practices.

Towards this goal, we established the Field Observatory Network (FiON), a network of researchers, farmers, companies and other stakeholders applying carbon farming practices. FiON has created a unified methodology to monitor and forecast agricultural carbon sequestration, by combining automated near real-time field measurements, weather data, satellite imagery, modeling and computing networks. In general, FiON follows the principles of other ecological observatory networks, such as National Ecological Observatory Network (NEON, Keller et al., 2008), Global Lake Ecological Observatory Network (GLEON, Hipsey et al., 2017) and Biodiversity Observatory Networks (GEOBON, Guerra et al., 2021) that collect long-term ecological data and monitor the effects of climate and land use change (Elmendorf 2016; Hinckley et al., 2016; Hipsey et al., 2017; Keller et al., 2008). The primary purpose of FiON, however, is to i) create new knowledge on soil processes, ii) to measure, verify and forecast the carbon sequestration in agricultural soils and to iii) approximate the effects of carbon farming practices on yield, biomass and $CO_2$ flux in near real-time. To achieve this, FiON invested in the use and development of a community cyberinfrastructure tool, Predictive Ecosystem Analyzer (PEcAn, pecanproject.org), which enables synthesizing different data sources and process-based models, quantifying and partitioning uncertainties, and operationalizing near real-time ecological forecasting (Fer et al., 2021). To disseminate the observations and findings, we built a free-access online dashboard called Field Observatory (v1.0, fieldobservatory.org). This website serves as a tool to monitor the impacts of carbon farming practices. The dashboard integrates data from field sensors, remote sensing and field survey. In this sense, FiON will provide decision support for the farmers, at first hand via the Field Observatory website and in due course via the scientific synthesis informed by the best available data and models. To serve the research and other interested communities, the data in Field Observatory is publicly available and downloadable from the website.

In this paper our objectives are to 1) describe data flows from various manual and automatic measurements in the Field observatory, 2) demonstrate 15-day forecasts of carbon exchange and plant growth towards decision support for the farmers, and 3) discuss the benefits of the public monitoring network established by FiON.

First, we introduce the sites included in FiON, and describe the tested carbon farming practices. Next, we describe the FiON workflow from data collection, processing and storage to visualization and dissemination through the Field Observatory website. Finally, we present the near real-time model-data synthesis, forecasting and decision support for the users.

## 2 Sites and tested carbon farming practices

The first phase of FiON consists of two intensive agricultural research sites and 20 voluntary farms testing carbon farming practices (Fig. 1, https://www.fieldobservatory.org/MapView). These 20 farms, called Advanced Carbon Action farms (ACA), were selected out of 100 pilot farms participating in the Carbon Action platform[1], where volunteer farmers test carbon farming practices (Mattila et al. 2022). Each farm has a test field and an adjacent, conventionally managed, control field (field 1 and 0 in Field Observatory, respectively). The additional carbon farming practices aim to increase carbon stock through increasing carbon inputs (photosynthesis & soil amendments) or through decreasing carbon decomposition (Minasny et al., 2017). These practices (Table 1) are: cover crops, adaptive grazing, soil amendments, subsoiling and ley farming (introducing a grass crop into rotation). Each farmer made a five-year carbon farming plan and took soil samples at the beginning of the study from GPS located points in the field. The same points are monitored annually and also contain real-time soil sensors.

---

[1]

Carbon Action platform consists of several scientific projects, 100 farms committed to 5 years of research activity and farmer extension services. As of spring 2021, some 600 farmers are participating around the topic. Food system companies and organisations are also involved. Carbon Action is led by BSAG and the research is coordinated by FMI. More https://carbonaction.org/en/front-page/

82

**Table 1 Principles of the carbon farming practices tested at the Carbon Action farms.**

| Carbon farming practice | Principles for carbon sequestration |
|---|---|
| Cover crops | Crops planted to lengthen photosynthetically active period and to increase carbon inputs from above and belowground biomass and to reduce leaching of carbon and nutrients. |
| Adaptive grazing | Short grazing & long rest periods to manage grass growth for increased root growth and increased soil cover. |
| Soil amendments | Exogenous carbon input. High input of organic material may stimulate plant growth through increased water holding capacity, nutrients, etc. |
| Subsoiling | Removing physical barriers to root growth by soil loosening. Coupled to a grass crop to stabilize loosened soil. Increases plant growth and soil aeration and decreases bulk density. |
| Ley farming | Breaking monocropping with perennial grass. Increases photosynthesis, root input and diversity. |
| Grass cultivation | Diverse plant species composition, increased cutting height and organic fertilization. |

84

The 20 ACA farms were selected based on their chosen practice (four farms per measure), location (appropriate distances for survey work and even spread over Finnish farmland) and soil type (a mix of clay and sandy soils) (Table 2). All of them were included in a soil quality survey in 2019 (Mattila, 2020). Farms with anomalous measurements or too large organic matter content or nutrient differences between the control and treatment plots in the initial phase of FiON were excluded from ACA farms. FiON includes two intensive research sites, Qvidja and Ruukki, which are operated by the Finnish Meteorological Institute (FMI). In Qvidja, carbon farming practices are tested in three different fields. In Ruukki, there are no carbon farming practices implemented at the moment. Both sites have eddy covariance towers which continuously monitor greenhouse gas fluxes and weather (see Sect. 3).

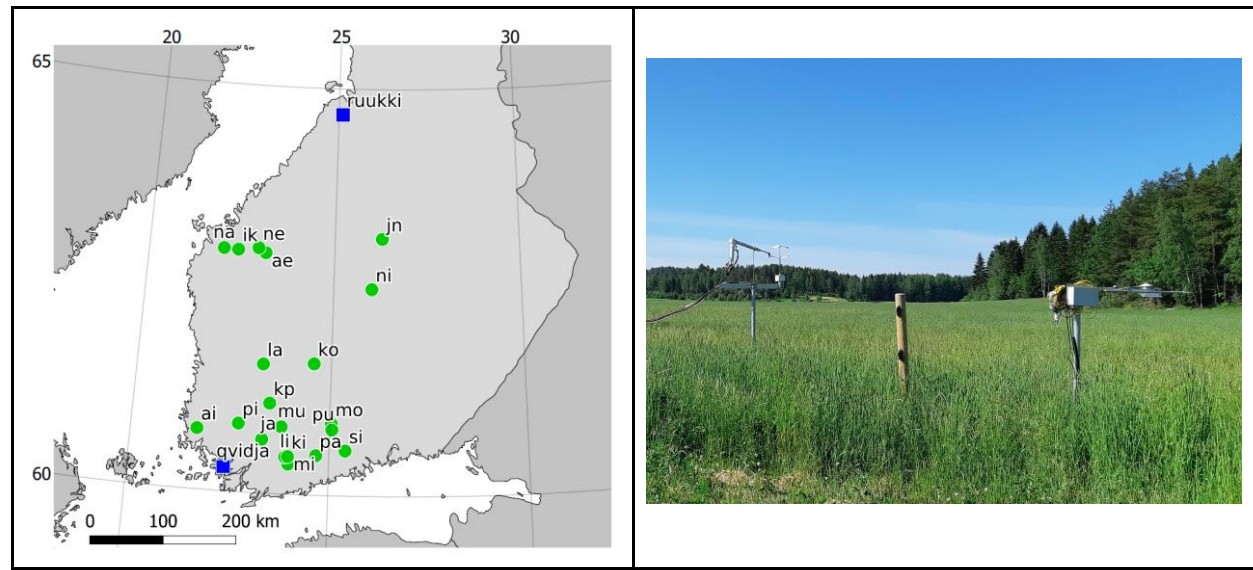

93

**Figure 1** Map of Advanced Carbon Action sites (green dots) and intensive sites (blue squares) (left), and eddy covariance
tower and radiation measurement instrumentation at Qvidja (right).

96

**Table 2 Current FiON sites.**

| Site | Site type | Soil type | Carbon farming practice | Species in 2020 | Nearest FMI weather station |
|------|-----------|-----------|--------------------------|------------------|------------------------------|
| AE | ACA | Sandy loam | Subsoiling | Rye | Kauhava airport |
| KO | ACA | Silt | Subsoiling | Silage grass | Juupajoki Hyytiälä |
| KP | ACA | Clay loam | Subsoiling | Multi-species ley | Pirkkala airport |
| LA | ACA | Clay silt | Subsoiling | Oats | Pirkkala airport |
| JN | ACA | Fine sand | Adaptive grazing | Pasture grass | Vesanto Sonkari |
| MI | ACA | Clay loam | Adaptive grazing | Pasture grass | Lohja Porla |
| NI | ACA | Sand till | Adaptive grazing | Pasture grass | Jyväskylä airport AWOS |
| KI | ACA | Fine sand | Soil amendments | Multi-species ley | Somero Salkola |
| LI | ACA | Clay loam | Soil amendments | Spring wheat | Lohja Porla |
| PA | ACA | Clay loam | Soil amendments | Hay grass | Nurmijärvi Röykkä |
| PI | ACA | Clay loam | Soil amendments | Oats | Kaarina Yltöinen |

| | | | | | |
|---|---|---|---|---|---|
| MU | ACA | Clay loam | Grass mixture | Multi-species ley | Somero Salkola |
| NA | ACA | Loam | Cover crops | Peas | Vaasa airport |
| NE | ACA | Loam | Cover crops | Oats | Kauhava airport |
| PU | ACA | Silty clay loam | Cover crops | Oats | Mäntsälä Hirvihaara |
| SI | ACA | Clay loam | Cover crops | Multi-species ley | Porvoo Harabacka |
| AI | ACA | Silty clay | Ley farming | Multi-species ley | Rauma Pyynpää |
| JA | ACA | Clay loam | Ley farming | Multi-species ley | Jokioinen Ilmala |
| IK | ACA | Sand till | Ley farming | Silage grass | Seinäjoki Pelmaa |
| MO | ACA | Loam | Ley farming | Barley | Hämeenlinna Lammi Pappila |
| Qvidja | Intensive | Clay loam | Grass cultivation | Silage grass | Kaarina Yltöinen[*] |
| Ruukki | Intensive | Organic (peat) | - | Silage grass | Siikajoki Ruukki[*] |

*Intensive sites have their own micrometeorological measurements.

**3 Data collection**

FiON combines multiple online and offline data streams with different temporal frequencies and geographical extent (Fig. 2, Table 3). These data streams flow into a server where the data are pre-processed (filtered, gap-filled, formatted) and model-data analyses are performed through an ecological cyberinfrastructure Predictive Ecosystem Analyzer (PEcAn, Fer et al., 2021). All observational and computational outputs are stored in the server and disseminated through a web-based user interface. In the following sections we describe each data stream and model-data activity in the order given in Fig. 2.

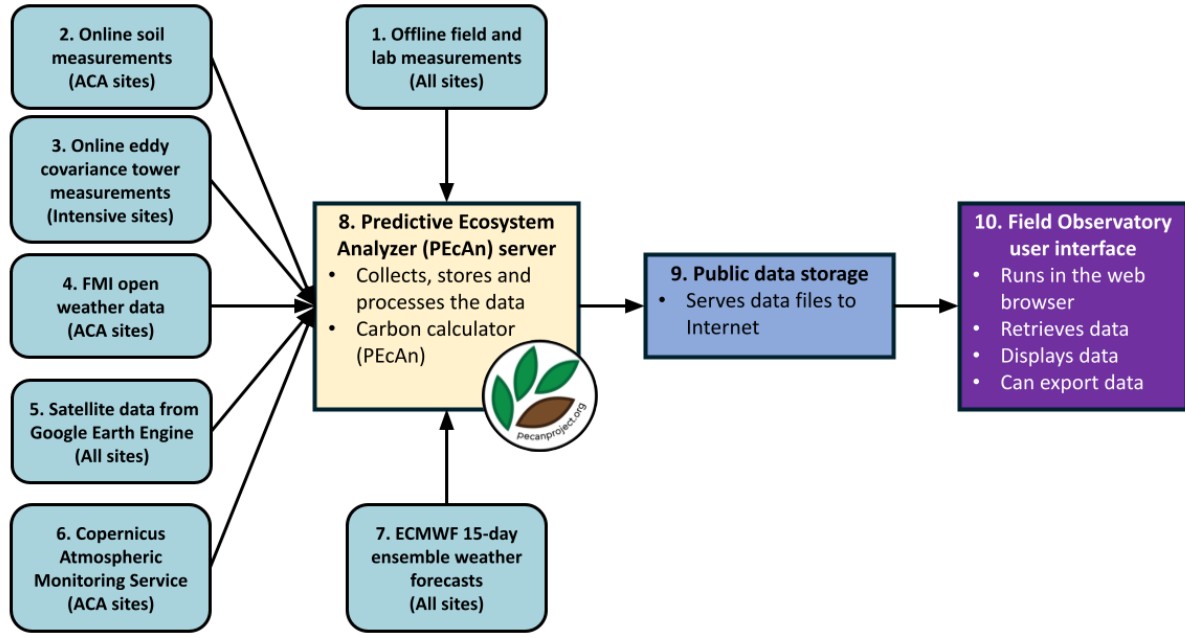

**Figure 2** Overview of the FiON data flows.

109

## 3.1 Offline field and lab measurements

At ACA sites, the measurements are done at three georeferenced points per field. The points have c.a. 30–100 m distance from each other and are located on a transect line. The transect was situated on each field to ensure comparable conditions for both the test and control plots. When placing the transects, slope, vegetation map and soil type were used to ensure the transect covers different management zones in the field. Annual soil sampling and soil quality measurements are made within a ten-meter radius of these points. All offline data from ACA sites on soil properties (cation exchange capacity, pH, organic matter), nutrients (P, K, S, Ca, Mg, Cu, Zn, B, Mn, Fe, Al, P-saturation), soil physical quality (soil structure, bulk density, porosity, water holding capacity, infiltration rate) and biological properties (earthworm counts, above ground biomass, percentage plant cover) are presented in Zenodo data repository with annual updates (Mattila, 2020; Mattila and Heinonen, 2021). In addition to annual monitoring, a pre-study SOC sampling was conducted on the fields in 2018 and will be repeated in 2023. In these studies, ten 20 cm deep core soil samples (14 mm diameter) were collected at 10 m radius from a georeferenced point centre and pooled to form a composite sample. Such samples were taken from each field from the three measurement points from both the control and carbon farming fields. Focusing the sampling to georeferenced locations and using composite sampling,

reduces the overall sampling variability and allows tracking relatively small (4 % of background level) changes in SOC stock

(Knebl et al. 2015). The offline field measurements at the intensive site Qvidja are described in Heimsch et al. (2021).

Offline, non-automated and infrequent data are currently being curated further for harmonization and reporting in JavaScript

Object Notation (JSON) file formats and International Consortium for Agricultural Systems Applications (ICASA) standards

(White et al., 2013). An example soil carbon measurement data point ($16.59 \pm 2.25$ kg m$^{-2}$, average $\pm$ standard deviation) is

visualized on Qvidja graphs and available on the accompanying JSON file (https://data.lit.fmi.fi/field-

observatory/qvidja/ec/events.json).

**3.1.1 Field activity**

All field activity information (e.g. planting, fertilization, harvest timing and amount) is currently received offline through

personal communication. An online application is under development for i) harmonizing historical field data, and for ii)

collecting future field activity data. Accordingly, the application is being developed to allow the farmers themselves to enter

these events and related details, and it will be tested for the first time at the end of the 2021 season. The application is written

using the Shiny R package (v1.6.0, Chang et al., 2021) and it automatically produces files in a JSON format using the ICASA

standards when possible (https://github.com/Ottis1/fo_management_data_input). Examples of historical field activity events

(e.g. planting and tillage) that are prepared through this application are being made available in the Field Observatory JSON

files and visualized on the graphs (Fig. 5).

**3.2 Online soil measurements**

Since 2020, each ACA site was provided with four TEROS-12 soil sensors (METER Group, Inc. USA) (two sensors per field,

control and treatment) measuring volumetric water content, electrical conductivity and temperature (Table 3). The automated

sensors are located at 75 mm depth in two of the three fixed measurement points of each field. The sensors were connected to

a third party data transfer hardware (Datasense Oy, Finland), which uses Lora/WAN network to transmit the data. During the

first year, the sensors measured every half hour but in 2021 measurement frequency was changed to one hour. The data is

stored at the service providers server and is pulled to the PEcAn server (#8) through the Datasense API. Currently the sensor

array includes 80 TEROS-12 soil sensors, four $O_2$ sensors (Apogee Instruments, SO-120, USA) and two $CO_2$ sensors (Vaisala

Oy, G525, Finland) and will be supplemented with weather and groundwater depth measurements.  The soil $O_2$ and $CO_2$ meters

are used to track changes in soil microbial activity and to guide model development.

**3.3 Online eddy covariance measurements**

Carbon dioxide, evapotranspiration (latent heat), sensible heat and momentum fluxes between the ecosystem and atmosphere

are measured at the intensive study sites, Ruukki and Qvidja, using the micrometeorological eddy covariance (EC) technique.

The EC instrumentation at both sites includes a three-axis sonic anemometer (uSonic-2 Scientific, METEK GmbH, Elmshorn,

Germany) and an enclosed-path infrared gas analyser (LI-7200, LI-COR Biosciences, NE, USA) installed on a tower. The measurement height is 2.3 m in Qvidja and 3.3 m in Ruukki (2.3 m from 13 June to 25 June 2019, 3.1 m from 25 June to 4 November 2019 and 3.3 m since 5 November 2019). The measurement heights fulfill guidelines for grasslands and croplands defined by the Integrated Carbon Observation System (ICOS; Sabbatini & Papale, 2017). For details of the measurement set-up in Qvidja, see Heimsch et al. (2021).

The data from the EC instruments are recorded at a 10-Hz frequency. Half-hourly turbulent fluxes are calculated by block-averaging these raw data after applying a double rotation of the coordinate system (McMillen, 1988). The time lag between the sonic anemometer and gas analyzer signals is determined based on the cross-correlation analysis (Rebmann et al., 2012). The gas fluxes are calculated from the mixing ratios determined with respect to dry air (Webb et al., 1980). The measured fluxes are compensated for the losses due to high-frequency signal attenuation within the measurement system (Laurila et al., 2005). The flux data are filtered for instrument malfunction and unfavourable flow conditions according to the following generic validity criteria: number of spikes in the raw data < 100, mean $CO_2$ mixing ratio > 350 ppm, relative stationarity (Foken and Wichura, 1996) < 30 % and $CO_2$ mixing ratio variance < 15 ppm² from April to September and < 5 ppm² from October to March. At the Ruukki site, flux data are accepted from the wind direction sector 135°-315° (Blocks 5, 6, 5up and 6up) and the sectors 0°-90° and 330°-360° (Blocks 1-4). In Qvidja, the wind directions representing the direction of the experimental site are 0°–30° and 140°–360°. Periods of weak turbulence are filtered by applying a site-specific friction velocity threshold. The threshold and its uncertainty are estimated for each site-year using the moving-point-transition method (Reichstein et al., 2005) and a bootstrapping approach (Pastorello et al., 2020). For incomplete years, the estimates from the previous year are used. While the flux data provided online are screened, they will be subject to further quality control in offline post-processing that will produce the final datasets distributed for scientific use. These post-processing procedures include flux footprint analysis and related data screening for inadequate upwind fetch, i.e., for cases in which the measured flux does not predominantly represent the field. Footprints are calculated with respect to the effective measurement height that takes into account the varying canopy height and snow depth.

The EC measurements are complemented with supporting meteorological observations conducted next to the flux tower. These include soil moisture, soil temperature at different depths, soil heat flux, photosynthetically active radiation (PAR), global and reflected solar radiation, air temperature and precipitation. Half-hourly meteorological and flux data are transmitted to a server at the FMI, which is then synchronized to the PEcAn server (#8).

### 3.3.1 Flux gap-filling and uncertainty analysis

The missing $CO_2$ flux (net ecosystem exchange, NEE) data are gap-filled based on empirical response functions that are fitted separately for the gross primary production (GPP) and total ecosystem respiration (ER):

$$NEE \ = \ GPP \ + \ ER \tag{1}$$

Respiration is modelled as a function of air temperature:

$ER = R_0 \cdot e^{E_0 \cdot \left( \frac{1}{T_0} - \frac{1}{T_a - T_1} \right)}$          (2)

where $R_0$ is the respiration rate at the reference temperature of 283.15 K, $T_0 = 227.13$ K, $T_1 = 56.02$ K, $E_0$ is the temperature

sensitivity of respiration, and $T_a$ is the measured air temperature (Lloyd and Taylor, 1994).

GPP is modelled as a function of PAR:

$GPP = \frac{\alpha \cdot PAR \cdot GP_{max}}{\alpha \cdot PAR + GP_{max}}$          (3)

where $\alpha$ is the apparent quantum yield and $GP_{max}$ is the asymptotic photosynthesis rate in optimal light conditions.

For gap-filling, the data are divided into sections based on the harvest dates, and each section is gap-filled separately. This is

done because fluxes measured before a harvest cannot be used to predict fluxes after a harvest. First, $R_0$ and $E_0$ are estimated

from the night-time (PAR < 20 µmol m$^{-2}$ s$^{-1}$ ) flux data with a 15-day moving window. If there are less than 25 observations,

the window size is increased stepwise by two days until enough data are obtained. Similarly, $\alpha$ and $GP_{max}$ are determined with

a three-day moving window by fitting the PAR response function to the daytime NEE from which the modelled respiration is

subtracted. Finally, gaps in NEE are filled with modelled NEE, which is the sum of modelled GPP and modelled ER. Gap-

filled values that are determined using fits from asymmetrical time windows, with possibly biased data are flagged and updated

when new measurements become available. Before flux gap-filling, the missing air temperature and PAR data are imputed

using linear interpolation if the gap is not longer than 6 h. Longer gaps are filled using the mean diel cycle of the data measured

within seven days before or after the missing data point

The uncertainty of measured NEE ($u_{meas}$) is inferred from the model residuals. For each site-year, the measurements are

grouped into 0.2 mg $CO_2$ m$^{-2}$ s$^{-1}$ wide bins, and for each bin the measurement uncertainty is characterized as the standard

deviation of the residuals. The uncertainty of each measured half-hourly flux is then estimated from the relation between the

measurement uncertainty and the magnitude of the flux (Richardson et al., 2008). For incomplete years, the relation from the

previous year is used.

The uncertainty of modelled NEE ($u_{mod}$), Eqs. (1)–(3), is propagated from the uncertainties of the least-squares fits of modelled

GPP ($u_{GPP}$) and Reco ($u_{Reco}$) as:

$u_{mod} = \sqrt{u_{GPP}{}^2 + u_{Reco}{}^2}$          (4)

Finally, the uncertainty related to the friction velocity threshold ($u_{ustar}$) is estimated by filtering the flux data using the 100

different bootstrapped friction velocity thresholds, gap-filling the 100 differently filtered datasets, and using the standard

deviation of the gap-filled fluxes as an estimate for $u_{ustar}$.

**3.4 FMI open weather data**

For all ACA sites, the weather information, namely precipitation, air temperature, relative humidity, wind speed and wind

direction are retrieved from the nearest FMI weather stations (Table 2). Weather data are pulled to the PEcAn server using the

fmir R package (https://github.com/mikmart/fmir).

**3.5 Satellite data from Google Earth Engine (GEE)**

All sites are monitored using remote sensing imagery from European Space Agency's (ESA) Sentinel-2 satellites.

Atmospherically corrected Level-2A (L2A) Sentinel-2 multispectral data (processed using Sen2Cor software) are retrieved

using GEE (earthengine.google.com) cloud data platform. The scene classification band available in L2A products is used to

filter away image acquisition dates during which the field is covered by snow, cloud or cloud shadow. From the Sentinel-2

data, we calculate the Normalized Difference Vegetation Index (NDVI) and the Leaf Area Index (LAI). LAI is calculated

because it is present in and can be assimilated to many process-based ecosystem models. NDVI is included in Field

Observatory mainly for the farmers to whom NDVI is a more familiar measure compared to LAI. NDVI is calculated using

near infra-red (B8A) and red (B4) bands of the L2A products. LAI is estimated using the ESA Sentinel Application Platform

(SNAP) Biophysical Processor neural network algorithm (Weiss & Baret, 2016,

https://github.com/ollinevalainen/satellitetools). The NDVI data is natively available in 10 m resolution, whereas LAI is

resampled to 10 m resolution from its original 20 m resolution. The satellite data is updated every two days at most (which is

the Sentinel-2 revisit frequency over Finland). In addition, yearly cumulative NDVI sum is calculated using integration by

trapezoidal rule for all sites ("NDVI days"). Common starting and ending points for the active growing season, 31 March and

31 October, respectively, are used to standardize the cumulative NDVI sums between sites. This standardization improves the

comparability of the cumulative sums between sites by having them all in the same absolute units. Without standardization the

cumulative sums would be much influenced by the availability of the first and last observations of the growing season for a

site. This is determined more by the cloud cover than the actual start and end of the growing season. To improve within site

comparison, the cumulative NDVI is computed using dates when all fields within a site have satellite imagery available. The

NDVI and LAI data is provided to the Field Observatory user interface in both raster (GeoTIFF) and tabular form (CSV).

With the tabular data, the average value of pixels within the field is used to estimate the field-level value. The tabular data is

provided with 90 % confidence intervals by multiplying the associated uncertainties by the Z-score for two-sided 90 %

confidence interval (1.645). Non-realistic negative LAI values are capped to zero. For NDVI the uncertainty is presented as

standard error of the mean (SE) of the pixels within the field. For the cumulative NDVI sum, the uncertainties are propagated

using the Python uncertainties package (https://pythonhosted.org/uncertainties/) which automatically computes the required derivatives and propagates the uncertainties.

The uncertainty for the LAI ($u_{LAI}$) is estimated by combining the observational uncertainty ($SE_{LAI}$) and the algorithmic uncertainty ($u_{alg}$) of the LAI estimation:

$$u_{LAI} = \sqrt{SE_{LAI}^2 + u_{alg}^2}, \tag{5}$$

where the $SE_{LAI}$ is computed as the SE of LAI observations within the field. The observational uncertainty aims at capturing the uncertainty associated with a particular single observation (from a specific image at a certain date). It is affected by the variability of the individual pixel values within the field at that specific date. The $u_{alg}$ is calculated by propagating theoretical individual pixel uncertainties ($u_{t_i}$) to the calculated average:

$$u_{alg} = n^{-1} \sqrt{\sum_{i=1}^{n} u_{t_i}^2}, \tag{6}$$

where n is the number of pixels (i.e. sample size) and $u_t$ the reported theoretical RMSE for the SNAP LAI algorithm that is 0.89 (Weiss and Baret, 2016) and constant to all pixels. The artificial increase of n due to resampling LAI observations from its native 20 m resolution to 10 m is taken into account and n is reduced accordingly.

**3.6 PAR from Copernicus Atmospheric Monitoring Service (CAMS)**

For the ACA sites, the daily PAR data are derived from the global irradiation data obtained from the CAMS through daily queries (www.soda-pro.com/web-services/radiation/cams-radiation-service/, Qu et al., 2017). The global daily irradiation (Wh m$^{-2}$ day$^{-1}$) is converted to daily PAR (MJ m$^{-2}$ day$^{-1}$) assuming that 50 % of the global irradiation is at PAR range. The CAMS data is available for each day with a 48 h time lag. The daily PAR is reported in MJ m$^{-2}$ day$^{-1}$ which is a more convenient unit for a daily value compared to µmol m$^{-2}$ s$^{-1}$ used with 30-min measurement frequency in intensive sites.

**3.7 ECMWF 15-day ensemble weather forecasts**

European Center Medium-range Weather Forecast (ECMWF) data are processed by the Finnish Meteorological Institute for every site. This dataset consists of 6-hourly 2 meter temperature (*2t* variable in ECMWF standards), total precipitation (*tp*), relative humidity (*r*), 10 meter U and V wind components (*10u* and *10v*, respectively), surface pressure (*sp*), surface solar and thermal radiation downwards (*ssrd* and *strd*, respectively) values of 51 ensemble members where one member is the control forecast and the other 50 are perturbed members which have perturbed initial conditions different than the control to explore

the range of uncertainty (Buizza and Richardson, 2017). Weather forecast data are updated everyday. Per ECMWF license agreements, the data are visualized as is but the disseminated tabular files are obfuscated.

**Table 3 Summary of data streams reported in FiON. Offline = stored in public data repository and updated as necessary.**

| Data type | Units | Data source | Frequency | Since | Sites | Online/offline |
|---|---|---|---|---|---|---|
| Field activity | - | Personal communication* | Seasonal | 2019 | All | Offline |
| Farmer management actions | - | Questionnaire | Annual | | All | Offline |
| Soil C | % (ACA), kg m$^{-2}$ (Qvidja) | Lab measurements | Biannual | 2018 | All, except Ruukki | Offline |
| Soil water holding capacity | m$^3$ m$^{-3}$ | Lab measurements | Once to calibrate sensors | 2019 | All, except Ruukki | Offline |
| Soil nutrients | mg kg$^{-1}$ | Lab measurements | Biannual | 2018 | ACA | Offline |
| Bulk density | kg dm$^{-3}$ | Lab measurements | Annual | 2019 | ACA | Offline |
| Biomass | kg ha$^{-1}$ | Lab measurements | Annual | 2019 | ACA | Offline |
| Soil moisture | m$^3$ m$^{-3}$ | ACA soil sensors & eddy covariance | Half-hourly | 2018 (Qvidja), 2019 (Ruukki), 2020 (ACA) | ACA & Intensive | Online |
| Soil temperature | °C | ACA soil sensors & eddy covariance | Half-hourly | 2018 (Qvidja), 2019 (Ruukki), 2020 (ACA) | ACA & Intensive | Online |
| Electrical conductivity | µS cm$^{-1}$ | ACA soil sensors | Half-hourly | 2020 | ACA | Online |
| $CO_2$-flux | mg m$^{-2}$ s$^{-1}$ | Eddy covariance | Half-hourly | 2018 (Qvidja), 2019 (Ruukki) | Intensive | Online |
| Latent and sensible heat flux | W m$^{-2}$ | Eddy covariance | Half-hourly | 2018 (Qvidja), 2019 (Ruukki) | Intensive | Online |
| Short-wave radiation (incoming and reflected) | W m$^{-2}$ | Eddy covariance | Half-hourly | 2018 (Qvidja), 2019 (Ruukki) | Intensive | Online |
| $CO_2$ concentration | ppm | Eddy covariance | Half-hourly | 2018 (Qvidja), 2019 (Ruukki) | Intensive | Online |
| Precipitation | mm | FMI open weather & eddy covariance | Half-hourly | 2018 (Qvidja), 2019 (ACA & Ruukki) | ACA & Intensive | Online |

| Air Temperature | °C | FMI open weather & eddy covariance | Half-hourly | 2018 (Qvidja), 2019 (ACA & Ruukki) | ACA & Intensive | Online |
|---|---|---|---|---|---|---|
| Relative Humidity | % | FMI open weather & eddy covariance | Half-hourly | 2018 (Qvidja), 2019 (ACA & Ruukki) | ACA & Intensive | Online |
| PAR | MJ m$^{-2}$ day$^{-1}$ µmol m$^{-2}$ s$^{-1}$ | Copernicus & eddy covariance | Daily & half-hourly | 2018 (Qvidja), 2019 (ACA & Ruukki) | ACA & Intensive | Online |
| Leaf Area Index | m$^2$ m$^{-2}$ | Sentinel-2, GEE | Min 2-days | 2018 (Qvidja), 2019 (ACA & Ruukki) | All | Online |
| NDVI | - | Sentinel-2, GEE | Min 2-days | 2018 (Qvidja), 2019 (ACA & Ruukki) | All | Online |

\*Online application is under development.

### 3.8 Predictive Ecosystem Analyzer (PEcAn) server

All FiON data are pooled in an FMI server where model-data integration cyberinfrastructure software PEcAn is installed and
compiled. PEcAn is an ecological informatics toolbox that consists of process-based models, a workflow management system
and analytical tools for model-data synthesis (LeBauer et al., 2013; Dietze et al., 2013). The automated PEcAn workflow calls
a series of modularized tasks that involve pre-processing of the model inputs, configuring and running the models, post-
processing model outputs and performing model-data integration analyses. Coupling a process-based model to this workflow
requires writing a model package which consists of a few interfacing scripts as PEcAn adopts intermediate input and output
file formats, and applies pre- and post-model run analyses to these standards (Fer et al., 2021). While there are already many
ecosystem models coupled to PEcAn and its design is general across process-based models, coupling of more models that can
simulate agricultural ecosystems is in progress. In this study, we coupled the BASGRA_N model (Basic Grassland Model,
Höglind et al., 2020) to the PEcAn workflow and demonstrated its use for the Qvidja site (see Sect. 4, Model-data synthesis).
In the future, we will provide model predictions for all sites through PEcAn.

### 3.9 Public data storage

To harmonize the data, all tabular data with less than daily measurement frequency is aggregated to a 30 min interval (to every
283 hour and half hour) before transferring the data to the public data storage (Amazon Simple Storage Service, field-
284 observatory.data.lit.fmi.fi). To protect the privacy of the farmers, all data holding spatial information is transformed for all
285 ACA sites, except for site MU (which is operated by Häme University of Applied Sciences).

**3.10 Field Observatory user interface**

The Field Observatory user interface (v1.0, fieldobservatory.org) allows viewing general information about the sites and the measurements and carbon farming practices conducted on them. The website has an interactive map to navigate to site-specific dashboards. A site view consists of general information about the site, an interactive map with satellite imagery of a specified vegetation parameter, an interactive timeline for selecting satellite imagery for viewing, and a panel of interactive time series charts (Fig. 3). Each chart comes with a description of the displayed data. A chart typically contains multiple time series and the visibility of each can be toggled. The user can enable and disable time aggregation and choose the time aggregation level from predefined options. The time aggregation is calculated using sliding statistics such as mean or sum depending on the data type. Any chart can be exported as an SVG image or as a CSV file containing the displayed data. A global specification file defines a list of charts and the data source types that can be shown in each chart. Site-specific specification files are used to define data source types available for each site and to provide links to the data files. Specification files are stored in JSON format.

The website is served by Azure services. The map and site views are based on client-side JavaScript, running in the user's web browser. Maps have been implemented using Mapbox GL JS JavaScript library.

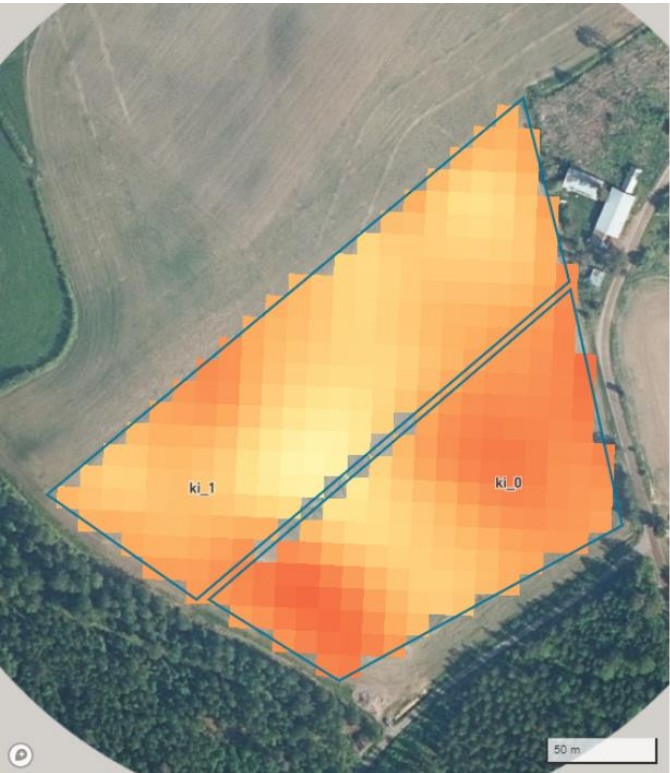

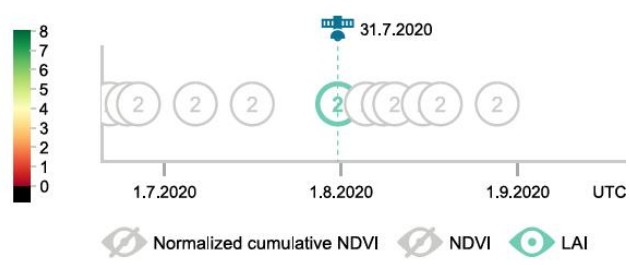

(a)

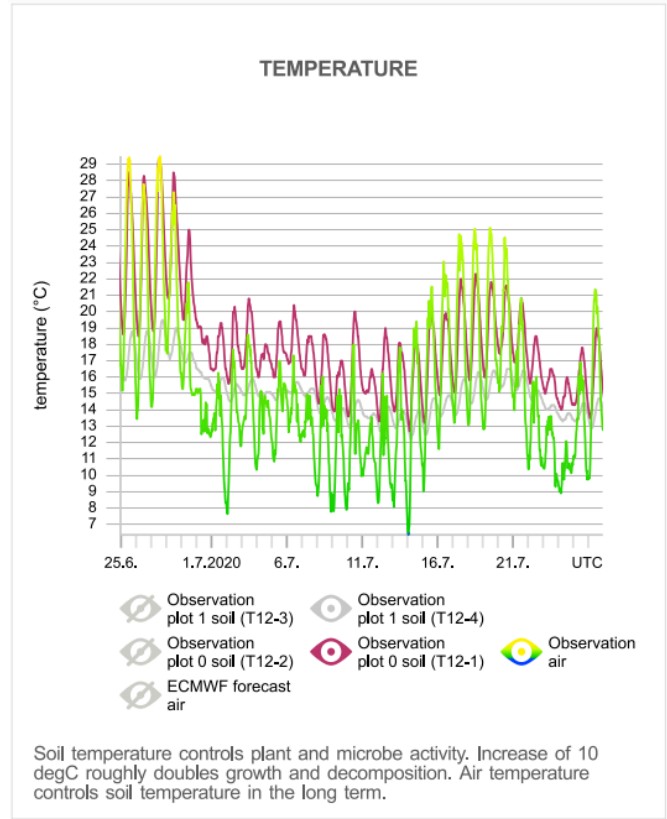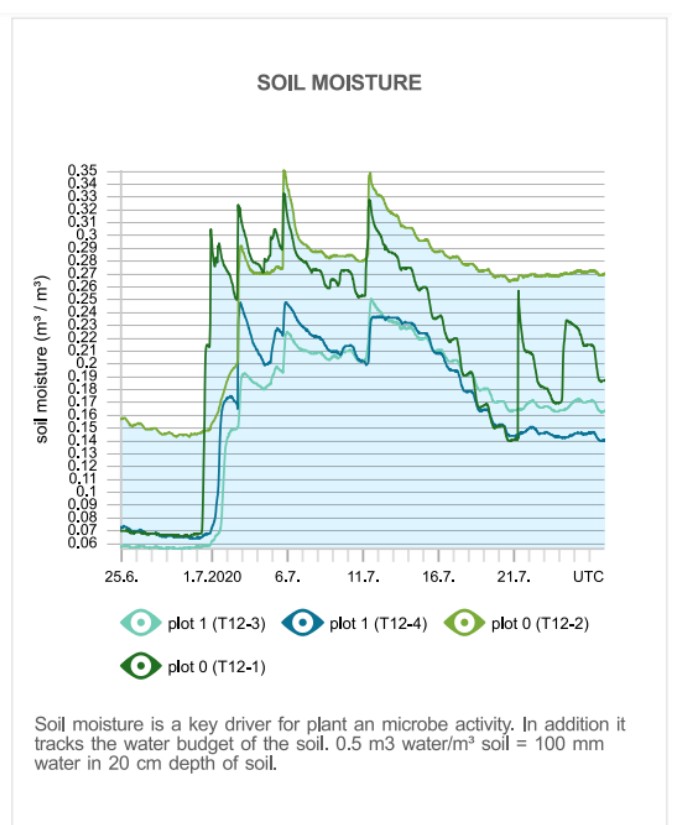

(b)

**Figure 3** Two web interface views of the measurement data for site KI: (a) Overview and LAI satellite images and (b) observed
soil and air temperature and soil moisture. The reader is referred to the website www.fieldobservatory.org for more and
interactive charts. The aerial photo contains data from the National Land Survey of Finland Topographic Database (11/2020).

**4 Model-data synthesis and decision support**

While the current version of the Field Observatory mainly disseminates observations, one of the main goals of this application
is to provide accessible near real-time model-data synthesis, forecasting and decision support for the users. We demonstrate

the first application of this service at the Qvidja grassland site with the grassland model BASGRA_N (Table 2). BASGRA_N model is developed specifically for northern climates and for grass types (timothy, *Phleum pratense*; meadow fescue, *Festuca pratensis*) that are the dominating forage species cultivated at the Qvidja farm, and it is able to simulate grassland productivity, quality and greenhouse gas balance (Höglind et al., 2020).

We coupled BASGRA_N to PEcAn, and used PEcAn's workflow management system and analytical tools (specifically the Bayesian calibration and state data assimilation modules) to inform the model with the data. Before employing them for forecasting and decision support, these models need to be initialized and calibrated. In other words, while state data assimilation algorithms can inform model states and improve predictive performance, best results are achieved when the model is calibrated to the site (Huang et al., 2021). Therefore, we used the field and lab measurements (Sect. 3.1), such as the rooting depth, soil carbon content and soil water holding capacity, to initialize the model states. Next, using multiple constraints ($CO_2$ flux and LAI from the eddy covariance tower field, Sect. 3.3), we calibrated 20 model parameters using Bayesian numerical methods through the BayesianTools R-package (Hartig et al., 2019) as implemented in the PEcAn system (Fer et al., 2018, also please see the supplement, section S1 for further details on the calibration protocol). In calibration, we used the observations from May 2018 to April 2021. After calibration model predictions were improved in terms of both uncertainty reduction and accuracy (Fig. 4). While the model is calibrated by the EC field data at Qvidja, initial results show improvement at the nearby Qvidja ACA sites as well (not shown here, but visible on Field Observatory LAI graphs).

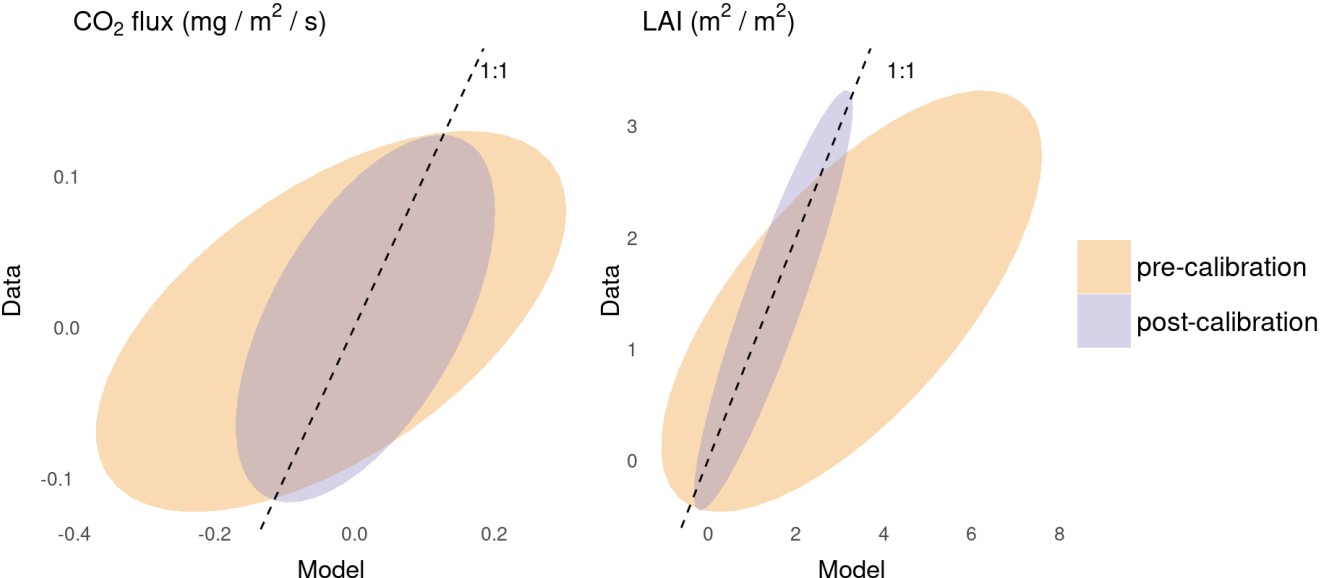

**Figure 4** Predicted versus observed comparison before (orange ellipses) and after (purple ellipses) initialization and calibration. Ellipses represent the 90 % confidence intervals of model ensemble runs with 500 members. After initialization

and calibration, the model performance at Qvidja improved in terms of both accuracy (closer to the 1:1 line) and uncertainty reduction (narrower ellipses).

Next, we deployed the initialized and calibrated model in an online, operational, iterative near-term forecasting framework by driving it with the ECMWF ensemble 15-day weather forecast (Sect. 3.7). From April 2021 onwards, every day a 15-day ensemble forecast is made from the BASGRA_N model. As time progresses, each day the $CO_2$ flux forecast is informed with the observed and gap-filled daily $CO_2$ flux values within an iterative forecast-analysis cycle using the Ensemble Adjustment Kalman filter algorithm implemented in PEcAn (Dietze, 2017). When LAI observations are also available, they are jointly assimilated with the $CO_2$ flux measurements as well. Although we are currently only assimilating the $CO_2$ flux and LAI observations, related states are also updated within the model through the analysis step as the model encodes and simulates relations and covariances among different ecosystem processes. Among the model output variables, we share the LAI and $CO_2$ flux (Fig. 5), as well as Latent Heat and Yield Potential forecasts with the users through the Field Observatory user interface, albeit only for the Qvidja site for the time being.

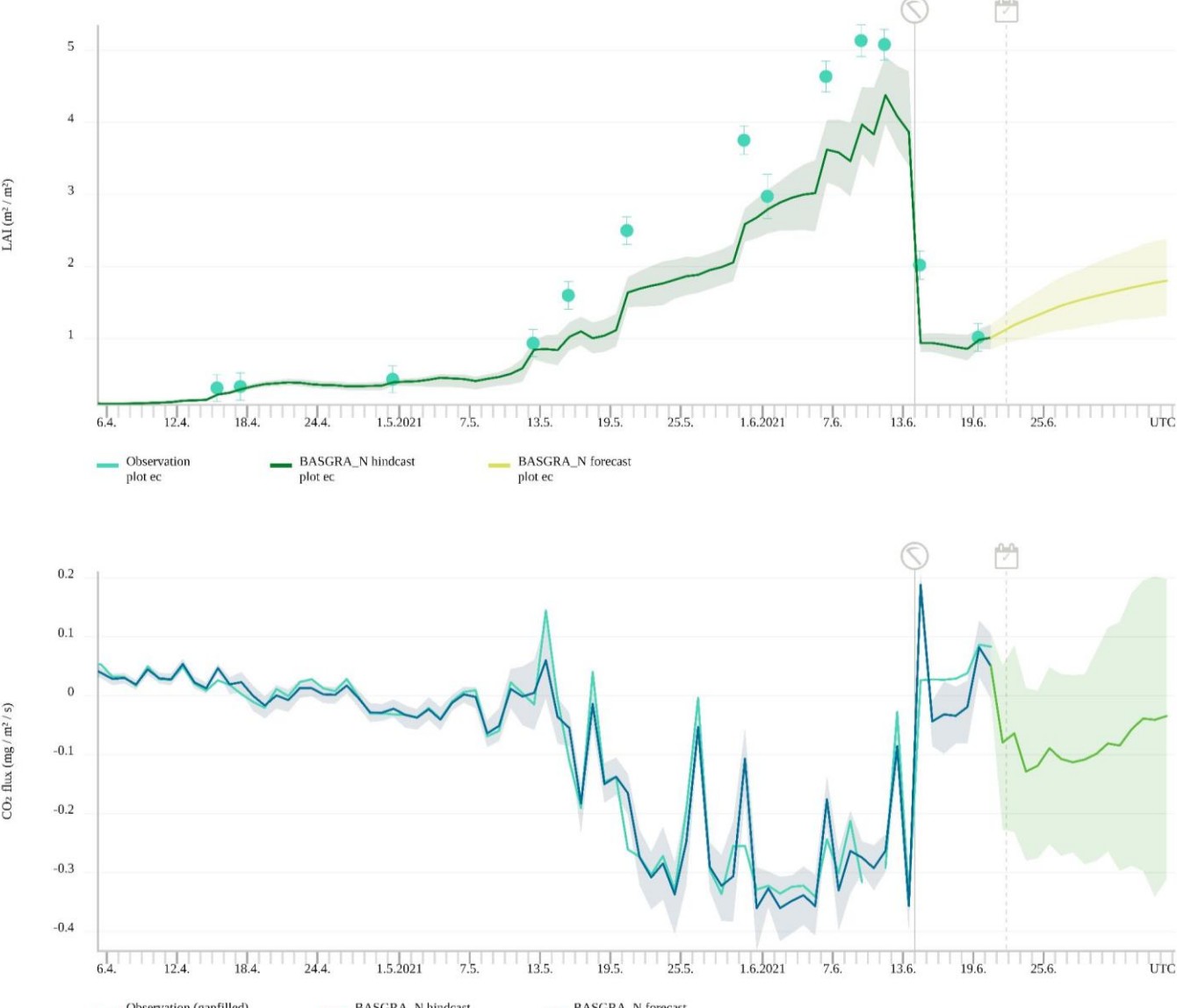

Figure 5 15-day LAI (top) and $CO_2$ flux (bottom) forecast at Qvidja. The 90 % confidence intervals for hindcast and forecast are generated by 250 ensemble members, with different combinations of model parameters, initial conditions and meteorological drivers. Units in the $CO_2$ flux graph are given per second to reflect the measurement frequency, however, observations were aggregated to daily time step here to align with the model predictions. The scythe icon indicates a harvest event on June 14th, 2021.

While a 15-day forecast has limited applicability within a cropping cycle, it could be informative on certain field activities that may have 1–2 weeks of flexibility which in return may have an impact on carbon balance. For instance, one can simulate alternative scenarios of timing of the harvest (e.g. whether to harvest now or delay it, please see supplementary material S2 for a demonstration). It is possible to retrospectively explore these cases systematically as both weather forecasts and model analysis states are archived in the Field Observatory's operational iterative forecasting system.

## 5 Discussion

This paper introduced the Field Observatory Network (FiON) and its unified methodology leading the way to monitor and forecast the functioning of agricultural ecosystems, geared towards verification of soil carbon sequestration. This methodology combines the existing spatially scattered measurements, modeling and computing networks, and disseminates the model-data computation outcomes through the Field Observatory user interface. In the following, we discuss the scientific and practical contributions of FiON and the Field Observatory, and the future steps planned for both.

### 5.1 Scientific contribution

FiON adopts state-of-the-art field and laboratory methods, open data sources, near real-time satellite imagery processing and model-data integration cyberinfrastructures—all of which are needed for a reliable MRV platform. A distinct feature of FiON is the network of ordinary farms, ACA sites, to establish baseline trends and verify additional changes. As soil carbon pool changes slowly, even after a shift in management practices, long-term monitoring is needed. The ACA sites (with control and treatment plots) were specifically designed for this purpose and will be monitored continuously for at least the next five years, and FiON aspires to continue even longer. This is an adequate time frame to detect SOC changes because the fastest carbon re-accumulation occurs in the first 10–20 years, depending on soil type, management practices, climate and initial SOC (Bossio et al., 2020), all of which are monitored by FiON. The intensive and ACA sites provide an important benchmarking opportunity to our model-data synthesis methodology which will be applied to all 100 Carbon Action farms.

The PEcAn platform is central to our methodology; it enables synthesizing different data sources and process-based models, managing observational and model uncertainties, and near real-time forecasting. It distinguishes FiON from observations-only approaches. In addition to potentially having practical relevance for improving carbon storage, near-term agricultural forecasting has benefits to basic carbon science. Data assimilation methods help dissect model behaviour and identify research needs (Viskari et al., 2020). For instance, variability patterns of the best parameter sets in time and space can be identified by studying model ensemble members with respect to the analysis states (i.e. our best understanding about the system) and may point to unaccounted processes in models and underlying sources driving variability. If we manage to account for these variabilities (e.g. adding covariates that explain temporal variability), we could also improve our capability to model the carbon sequestration on the long term. Moreover, near-term iterative forecasting provides an out-of-sample way of statistical testing

for models which is less prone to overfitting than in-sample tests which are more typical in (agro-)ecology where models are tested against data that has already been observed (Dietze et al., 2018). Accordingly, a more in-depth analysis of the archived Field Observatory forecasting results and skills are ongoing and will be reported in a future study. In addition to understanding models better, operational iterative near-term forecasting also allows us to detect and intervene when measurements of certain sensors or data streams deviate from the assimilated background, and in return supports the management of the sensors and data pipelines, resulting in higher quality datasets. Overall, our 15-day iterative forecasting system provides continuous quantitative benchmarking of models and data based on all other available information, which allows rapid detection and explanation of changing patterns in the carbon sequestration with the possibility of intervening and making adjustments.

**5.2 Practical contribution**

The Field Observatory user interface has not only enabled farmers to monitor impacts of their carbon farming practices, but also to connect and compare their own and others' data and practices. Features in the user interface are co-created with the farmers and developed accordingly. For example, farmers requested to see a cumulative sum of NDVI through the growing season which was in return calculated and included on the website. Likewise, simple and clear descriptions to interpret each data type have been found helpful. The gap-filled $CO_2$ fluxes at the intensive study sites have made it easier to communicate carbon exchanges between land and the atmosphere and how carbon budget calculations are done. As a result, the Field Observatory has already been used in workshops and meetings with stakeholders, and in training and scientific outreach for the Carbon Action farmers.

One of our aims with this framework is to provide decision support for the end users. This is effectively offered by Field Observatory in terms of feedback where end users can monitor the impact of their activities in a quantitative manner, assess and make their decisions in the future accordingly. Our framework also lays the groundwork for a more explicit and specific decision support system. Although such functionality is not fully in place yet (but under development), establishing the operational data assimilation and iterative forecasting pipeline is a milestone towards this direction. While the 15 days' horizon has limitations with respect to the span of a production cycle, in the future we are planning to include seasonal, annual and longer term forecasts as well. However, 15-day forecasts can still provide decision support for relatively shorter term and flexible agricultural actions (such as harvest, irrigation, grazing etc.). With the additional layer of agricultural forecast on top of the weather forecasting services, users are quantitatively informed about the progression of various ecosystem states and services through these Field Observatory near-term forecast updates. For example, sensor or model-based dynamic fertilization strategies have successfully improved the nitrogen use efficiency of cropping systems (e.g. Sela et al., 2018; Scharf et al., 2011). Likewise, timing of harvest and the cutting height may affect the overall carbon budget and economic income, and the plants' water demand may necessitate a different irrigation scheme for optimum growth and water usage, all of which may not readily manifest themselves through weather forecasts and observations only. We also acknowledge that such interventions

are potentially easier for grasslands, as opposed to the croplands. Nevertheless, our operative iterative near-term forecasting system enables a framework to explore the impacts of such interventions dynamically, systematically and quantitatively, and in return devise more reliable and comprehensive decision criteria. Overall, the current pipeline is being developed to improve the model performance and to be put into an adaptive decision making framework where alternative scenarios will be simulated with the models to aid users in their day-to-day operations specific to their management structure and goals.

The near-term carbon forecasts have also improved our communication with stakeholders in general. Reporting quantitative, specific and iterative carbon forecasts makes it possible to convey the idea that predictive carbon science has the potential to be as successful and common as numerical weather prediction (NWP) as a discipline and as a service to society one day. Ecological forecasts provide us with a standard, quantitative, intuitive and management-relevant method and language to develop the right context and tools for structuring soil carbon sequestration decisions (Petchey et al., 2015; Dietze et al., 2018). Bringing near-term carbon forecasts forward further helps describe that soil carbon monitoring and modeling is a complex computational problem that depends on vast amounts of basic scientific research and observations. It involves a diverse range of actors and organizations and requires efficient communication and continuous transfer of knowledge between these groups, similar to NWP (Bauer et al., 2015). Not only the similarities but also the differences between agricultural forecasting and NWP help clarify and re-focus the research needs (e.g. the need to address the heterogeneity and inherent variability in carbon systems). Overall, near-term forecasts help establish this constructive dynamic between researchers and stakeholders which in return helps tackle remaining bottlenecks for improving soil carbon sequestration more efficiently.

There is a large interest towards adopting and developing Field Observatory further. Therefore, the website is under constant development with new features, and new information about carbon farming and findings of FiON are increasingly being made available.

**5.3 Avenues for future research and development**

We have planned future steps for both FiON and the Field Observatory. The first step is to add more agricultural models to PEcAn. This enables us to extend model-data analysis to all FiON sites where different species and management practices are involved (i.e. other than grass harvest timing and amount). Coupling of one such additional model (Simulateur mulTIdisciplinaire pour les Cultures Standard, STICS, Brisson et al., 1998) to PEcAn has already been completed, and others are in progress. In the meantime, more sites will be added to FiON, not only in number but also in type. For example, with carbon-smart planning, urban vegetation also has potential to store more carbon. We study this also in FiON and consequently urban sites will be added. Another goal is to include forests and peatlands in FiON, which requires incorporating new process-based models in the FiON workflow. During the coming years, more field and laboratory measurement data will be collected and used to validate the model estimates and re-calibrate the models.

The framework designed by FiON and described in this paper provides the necessary mechanics to study the applicability and reliability of the models to simulate components of the carbon budget virtually in every field. While scalability has been the core idea for the design of this framework since the beginning, putting it to practical test is the main scientific next step. Currently, a factorial experimental design and simulation is ongoing where the performance of the models will be tested at multiple sites by informing them with various data streams. For this, we will start with constraints that can be made available virtually from everywhere and test which combinations, if any, can inform models enough to capture local carbon budget dynamics and components. Such constraints are for example LAI derived from remote sensing, soil moisture provided by inexpensive in-situ sensors, soil properties estimated from global products and yield. In this setup, the information contributed by the sites that are equipped with EC-towers will also be tested. For example, we will perform a factorial experiment at the ACA sites with and without the models being constrained by EC-data at the intensive sites. As we have additional data streams other than the mentioned constraint data types (e.g. biomass and soil C, Table 2) from ACA sites for evaluation, the framework described in this paper provides the means to carry out such multi-site in-depth analyses.

The development of the online application to gather field activity data from farmers is also in progress. The main purpose of this application is to make collection and utilization of field activity data in visualization and model-data synthesis pipelines easy. In this context, Field Observatory's interoperability with commercial farm management information systems needs to be studied in order to reduce the number of times farmers are filling out such information. An additional future use of this online application is planned to enable the farmers to simulate a predefined number of scenarios regarding their day-to-day operations by triggering automated PEcAn workflows—for example, given the next 15-day forecast, they will be able to optimize the timing and amounts of their field activity. We are also considering utilizing this online application for additional purposes: a) for compiling information from farmers regarding the flexibilities of their activities as this brings an additional practical constraint on the development of the model-based decision support system, b) for enabling new users to submit electronic requests and information about their fields to be part of the FiON network, c) for supporting peer-to-peer learning between farmers (Mattila et al., 2022).

We are currently also investigating the use of satellite data sources other than Sentinel-2 in retrieving information on vegetation and soil properties. In addition to satellite imagery, drones could be used as a source of remote sensing data. The current Sentinel-2 data filtering is based on the cloud detection available in the L2A products. This filtering approach has produced quite clean time series; some sites do not have any outliers and some have at the maximum one or two per year. The benefit of our methodology—where we assimilate observations as state variables to process-based models—is that single outliers, with optimally larger uncertainties, do not have too drastic of an effect on the model predictions. Nevertheless, we will continue to follow the performance of the filtering approach and improve it if necessary. Finally, the data streams used in data assimilation (to inform and update forecasts) will be increased and improvement in forecasting skills will be analyzed.

## 6 Conclusions

The Field Observatory Network (FiON) introduced in this paper is primarily a network of researchers, farmers, companies and other stakeholders developing carbon farming practices. FiON provides a unified methodology to monitor and forecast agricultural carbon sequestration by combining offline and near real-time field measurements, weather data, satellite imagery, modeling and computing networks. FiON disseminates data through the Field Observatory user interface (www.fieldobservatory.org). For farmers, FiON serves as a monitoring and decision support tool. In contrast to the mainstream decision support tools, FiON also provides the farmers access to other carbon farmers' data in the network. This enables comparisons and knowledge transfer between the carbon farmers.

FiON has several analogies to other ecological observatory networks, but unlike these existing networks, FiON is designed to provide near real-time information and forecasts concerning the carbon farming practices and to facilitate monitoring and verification of carbon sequestration. In this sense, FiON takes several steps forward from the mainstream of the ecological observatory networks known so far.

## 7 Data availability

The data displayed in the Field Observatory are available from the Field Observatory website (www.fieldobservatory.org) and from Amazon Simple Storage Service at https://field-observatory.data.lit.fmi.fi/index.html. Field measurements conducted at ACA sites in 2019 and 2020 are available from Zenodo data repository (Mattila, 2020: Mattila and Heinonen, 2021).

## 8 Code availability

The satellite data processing codes are available from a public GitHub repository (https://github.com/ollinevalainen/satellitetools). All PEcAn code is available openly on a GitHub repository (https://github.com/PecanProject/pecan). Field Activity data collection and curation application code which is under development is also available via GitHub (https://github.com/Ottis1/fo_management_data_input). Rest of the codes by the authors are not yet openly available.

## 9 Author contribution

Conceptualization, ON, ONi, IF, AJ, TM, OK, JK, LH, LM, PJ, LK, ÅS, AL, JHe, IK, JL; Data curation, ON, IF, ONi, TM, OKu; Formal Analysis, ON, IF, ONi, TM, LHe, HV, SG, TV, JV, JT, Funding acquisition, TM, LK, AL, TL, JHe, TA, IK, JL; Investigation, ON, IF, ONi, TM, LHe, HV, SG, TV, JV, JT; Methodology, ON, IF, ONi, TM, HV, LK, OKu, TV, JV, JT, JHe,

TA, JL; Project administration, TM, JK, LH, LK, ÅS, AL, TL, JHe, TA, IK, JL; Software, ON, ONi, IF, AJ, OK, OKu, HV, TV, JV, JT; Visualization, ON, ONi, IF, AJ, LM, PJ, OKu and comments from all; Writing – original draft preparation, all authors; Writing – review & editing, all authors.

**10 Competing interests**

The authors declare that they have no conflict of interest.

**11 Acknowledgments**

The work of HAMK has been conducted within the research project: *Carbon 4.0 - Analysis and utilization of biological data in complex carbon ecosystems* funded by the Ministry of Education and Culture of Finland [grant OKM/189/523/2018]. The work by FMI was supported by Business Finland [grant 6905/31/2018], The Strategic Research Council at the Academy of Finland [decision no 327214], the Academy of Finland Flagship Program [decision no 337552], the Ministry of Agriculture and Forestry of Finland [grant VN/5094/2021] and Maj and Tor Nessling foundation (grant 202000391). The work by SYKE was supported by The Strategic Research Council at the Academy of Finland [decision no 327350].

The authors want to thank the 20 farmers who allowed testing the framework on their Carbon Action fields. We also thank the owner of Ruukki farm, Natural Resources Institute Finland (Luke), and their employees for making it possible to have a measurement site there. In addition, we are thankful for the owners and staff of Qvidja farm.

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
