# Peer review of "Towards agricultural soil carbon monitoring, reporting and"

_Geoscientific Instrumentation, Methods and Data Systems, 2021_

## Author Comment (AC1)

**GI-2021-21 Reply to Reviewers**

We would like to thank both Reviewers for the valuable and constructive comments. Those are much appreciated. We have taken them into account and we will revise and improve the manuscript accordingly. Below you can find our replies to both reviews.

On behalf of all authors,
Olli Nevalainen

**Review 1**

*The manuscript by Nevalainen et al. introduces the Field Observatory Network and its methodology which combines data from different sources (offline and near real time in situ data, satellite imagery and climatic data) with ecosystem modelling to support the monitoring and verification of soil carbon sequestration in agricultural soils. The network currently consists of 20 pilot farm sites in Finland with the involvement of different actors from researchers to farmers with the common goal of optimizing agricultural practices in order to preserve and possibly enhance soil carbon stocks. Both data and simulations results are disseminated publicly through a web graphical user interface. Such a system and intended as a decision support for the farmers who are able to monitor the effects of agricultural practices in relation to crop productivity and carbon sequestration is innovative, however the limits of the decision-making based on the information currently provided by the FiON are lacking in the discussion section. In addition, some aspects of $CO_2$ flux data processing have to be clarified to make the current methodology scalable in view of an expansion of the sites network.*

*Overall, the paper is well structured, written in a clear and concise style and definitely within the scope of the journal GI. I advise that the paper is published after minor review based on the following remarks:*

Thank you for the comments and improvement suggestions. The main concerns regarding the lack of discussion about the limits of current decision support provided by FiON and clarifications about the flux data processing are addressed below in the specific comments.

*P1 L29: a MRV tool*

We respectfully disagree on this. The pronunciation of the acronym MRV starts with a vowel "E" and thus the indefinite article should be "an".

*P8, L52 "unfavourable flow conditions according to the following validity criteria:..". CO$_2$ flux data are filtered according to validity ranges, which are site specific, except for those resulting from the stationarity tests and u-star filtering. However, in the perspective of expanding the EC flux observation network within the FiON, the same threshold values (eg. CO$_2$ mixing ratio mean and variance) might not be applicable to guarantee the quality of the processed datasets of other sites. Therefore, it would be important to explain how these thresholds were determined, also referring to current methodological standards of other flux networks, when possible. Moreover, was a flux footprint analysis carried out in order to exclude flux data (partly) not targeting the monitored crops? If yes, the Authors should include the description of this processing step in section 3.3. Was the diurnal footprint always encompassed within the crop fields borders? Was the increase in the crop height accounted for in the displacement height used in the footprint model?*

The limits for the number of spikes in the sonic anemometer raw data and for the CO2 mixing ratio are generic criteria that were introduced to ensure that the instruments function properly and to discard disturbed measurements. These limits are not considered site-specific. In contrast to the Reviewer's comment, the limit employed in the u-star filtering is determined separately for each site, as explained in the manuscript (P9 L57-59). The methodology is consistent with the data processing procedures applied within the FLUXNET (Pastorello et al., 2020). This will be clarified in the revised manuscript.

While the flux data provided online are screened and gap-filled, they will be subject to further quality control. Flux footprint analyses are not carried out online but are included in the post-processing that will produce the final datasets distributed for scientific use. The footprint analysis is used to screen the data for cases in which the measured flux does not predominantly originate from the target area, i.e. within the field borders. The changes in the effective measurement height (measurement height minus displacement height) were accounted for by including crop height and snow depth data. These matters will be clarified in the revised manuscript.

*P11 L18 " ..by combining the observational uncertainty..*

Thank you, we will fix this in the revised manuscript.

*P20, L50 "One of our aims with this framework is to provide decision support for end users"*

*This sentence is key for the discussion section and for the message delivered by the paper. The Authors should discuss explicitly the limits of the provided decision support for farm management in terms of potential array of options given the current forecast window of 15 days. Consider that many factors of agricultural management are fixed within a production cycle (eg. crop type) and that many agricultural operations depend*

*strictly on the crop growing stages and have limited margins to be temporally shifted. Also, underline in the discussion what can the FiON decision support system provide in addition to the traditional weather forecasting services upon which farmers usually rely*

We thank the Reviewer for this comment. We agree that the current forecast window of 15 days has limitations with respect to the span of a production cycle. While in the future we are hoping to include seasonal, annual and longer term forecasts as well, we believe 15 days forecasts can still provide decision support for relatively shorter term agricultural actions such as harvest, irrigation, fertilization, grazing etc. as we now elaborate per other Reviewer's comments (please see discussion and a case study below regarding the usefulness of 15 days forecasts). Adding the additional layer of agricultural forecast on top of the weather forecasting services can reinforce traditional operations with process understanding. For example, sensor or model-based dynamic fertilization strategies have been successful in improving the nitrogen use efficiency of cropping systems (e.g. Sela et al., 2018; Scharf et al., 2011). Likewise, timing of harvest and the cutting height may affect the overall carbon budget and economic income, and the plant's water demand may necessitate a different irrigation scheme for optimum growth and water usage, all of which may not readily manifest themselves to farmers through weather forecasts only. We also acknowledge that such interventions are potentially easier for grasslands, compared to the cereals for instance. This is a good point and we will elaborate on this more in the discussion section of the revised manuscript accordingly. Finally, we also note that while the near term forecasts are an important element in providing decision support, Field Observatory also offers decision support in terms of "feedback" to the farmers where they can monitor the impact of their activities in a quantitative manner and make their decisions in the future accordingly. We will make sure to clarify this as well.

*P20, L73-76: "field activities". These activities are only related to the work of the farmers in the fields and should be better indicated with the tem "agricultural activities"*

We thank the Reviewer for this suggestion, but in our opinion the term "field activity" refers better to specifically in-field activities. In addition, the term field activity is a commonly used term in precision agriculture (e.g. Corrado et al., 2018 and https://www.agricircle.com/en/smart-farming/field-monitoring/). We also collect measurement events that in our opinion fits better under the term "field activity".

**Review 2**

The paper "Towards agricultural soil carbon monitoring, reporting and verification through Field Observatory Network (FiON)" presents the Field Observatory Network (FiON) that aims at establishing a unified methodology towards monitoring and forecasting of agricultural carbon sequestration by combining offline and near real-time field measurements, weather data, satellite imagery, modeling and computing networks.

General comments:

The paper is overall well written and some components of the observatory such as the multi-actor approach and the online service for near real-time model-data synthesis and decision support for the farmers are very valuable. However, several aspects of the methodology needs to be clarified and I didn't really see the added value of the forecasting system for improving carbon storage as the forecasting methodology is limited to 15 days.

We thank the Reviewer for this comment, however, we respectfully disagree about the usefulness of 15 day forecasts. While we acknowledge that some of these aspects would not necessarily improve carbon storage directly, we specifically value the following three points when it comes to 15-days forecasting system for improving predictive carbon science in general:

**1) It helps researchers understand and improve their theories and methods:** In addition to potentially having practical relevance for improving carbon storage (please see the 3$^{rd}$ point), near term agricultural forecasting has benefits to basic carbon science. Although this is not specific to carbon cycle models, data assimilation methods help dissect model behaviour and identify research needs (Viskari et al., 2020). For instance, a quick post hoc inspection where we evaluated the performance of ensemble members with respect to the analysis states (i.e. taking the analysis state and covariance matrix - our most accurate understanding about the system - and calculating the multivariate normal likelihood of the forecasted states with respect to the analysis) shows that while some ensemble members consistently perform well/worse, different ensemble members perform differently every time step, suggesting that optimal parameters may vary through time (Fig. 1.). Such parameter variability patterns in time may point to unaccounted processes in models and underlying sources driving variability. If we manage to account for these variabilities (e.g. adding covariates that explain temporal variability) we could also improve our capability to model the C sequestration on the long term. Furthermore, making out-of-sample predictions for a yet-to-be-observed future in a near-term iterative forecasting cycle provides a stronger test than in-sample statistical testing of our models and is less susceptible to overfitting (Dietze et al., 2018). Our forecast results are archived and a more in-depth analysis of the forecasting results and forecasting skills are ongoing and will be reported in a future study.

[Figure]

Fig. 1. Relative performances of randomly chosen (10) ensemble members at (20) different assimilation steps. 1: performing better, 0: performing worse. Some ensemble members perform consistently well (e.g. ens238) or worse (e.g. ens186) while different ensemble members perform differently each time step.

Moreover, in addition to understanding models better, operational iterative near-term forecasting also allows us to detect and intervene when measurements of certain sensors or data streams deviates from the assimilated background, and in return supports the management and maintenance of the sensors and data pipelines.

Overall, our 15-day iterative forecasting system provides continuous quantitative benchmark of models and data based on all other available information which is an essential element of reliable monitoring and modeling of the carbon cycle allowing rapid detection and explanation of changing patterns in the carbon sequestration, and possibly allowing them to be adjusted.

**2) It helps researchers relate carbon science to numerical weather prediction (NWP) and communicate better with stakeholders:** While clearly this does not have a direct impact on soil carbon in particular, reporting near-term forecasts of carbon improved our communication with stakeholders in general. Reporting quantitative, specific and iterative C forecasts makes it possible to convey the idea that this approach has the potential to be as successful and common as NWP as a discipline and as a service to society one day. Ecological forecasts provide us with a standard, quantitative, intuitive and management-relevant method and language to develop the right context and tools for structuring soil carbon sequestration decisions: facilitating that researchers contemplate a more stakeholder-orientated approach and stakeholders take a more active role in identifying appropriate variables and appropriate measures of proficiency given the

management structure and goals (Petchey et al., 2015).

Bringing near-term carbon forecasts forward further helps describe that soil carbon monitoring and modeling is a complex computational problem that depends on vast amounts of basic scientific research and observations, which involves a diverse range of actors and organizations and require efficient communication and continuous transfer of knowledge between these groups, similar to NWP. Not only the similarities but also the differences between agricultural forecasting and NWP help clarify and re-focus the research needs (e.g. the need to address the heterogeneity and inherent variability in carbon systems). Overall, near-term forecasts help establish this constructive dynamic between researchers and stakeholders which in return helps tackle remaining bottlenecks for improving soil carbon sequestration more efficiently.

**3) It provides decision support on near term events that can potentially have long-term effects on carbon storage:** Last but not least, while 15-days forecast has limited applicability within a cropping cycle, it could be informative on certain field activities that may have 1-2 weeks of flexibility which in return may have an impact on carbon balance.

**Case study:** For instance, the grass at the Qvidja farm had a remarkable difficulty in recovering after the first harvest of this year (14.6.2021) because the following days were particularly hot and dry. This resulted in poor carbon assimilation in the following weeks after the harvest. To devise a scenario where we evaluate the impact of timing of this harvest (e.g. what if it was harvested 1 week later), we travel back in time to 14.6.2021 where the farmer is facing the decision whether to cut the grass today (14.6.2021, what happened in reality) or later (e.g. 21.6.2021, alternative scenario). We restart the model from the states as of 13.6.2021 (which we archived) and run it forward with the 15-days weather forecast drivers available to us at the time (which we also archived). In other words, we start the model from its retrospective states and we are not using the now observed weather drivers for this period but the 15-day weather forecast in this exercise to mimic a real-life scenario. The available forecast horizon for the farmer in this case is the next 15 days as of 14.6.2021 (until and including 28.6.2021). We report the yield, LAI and NEE for the two scenarios where in one grass is cut on 14.6 and on 21.6 in the other:

[Figure]

Fig 2. Alternative scenario of delaying (right column) the harvest date (red vertical lines) leads to a slightly larger C sink within the 15-day period while resulting in less yield.

Delaying the harvest results in ~14% less yield for the farmer (Fig 2, top row). Both LAI development (Fig 2, middle row) and C exchange (Fig 2, bottom row) during the week following the harvest imply that delaying the harvest may have some adverse effects compared to the real case. While leaving the grass intact longer during the 14.6-21.6 period contributes to the overall C sink within this 2-weeks period (+0.08 vs alternative -1.16), it is not clear whether the amount (even if we account for yield difference as a C output from the system) justifies delaying harvest on the whole at the expense of yield. Looking at this picture, considering the next 15-days forecast on 14.6.2021, the farmer could be advised to keep the harvest date as 14.6 in this case study.

To observe post-hoc what will happen to the C sequestration at this period later in the season, we run the model forward until the end of our predefined growing season, 31.10 (note that this is just a forward run without being informed by the observations during this period, therefore should be taken as a relative comparison) and calculated the C-balance over the 14.6-31.10 period. When we consider only the fluxes between the ecosystem and atmosphere, the actual harvest time was more favourable, i.e. it acted as a higher sink than the alternative case (-1251 kgC ha-1 vs. alternative -861 kgC ha-1, Fig. 3). If the carbon removed as yield is also considered, the actual harvest day was again more favourable

(+288 kgC ha-1 vs. alternative +453 kgC ha-1, Fig. 3). This suggests that keeping the harvest time as it is might have indeed been preferred.

[Figure]

Fig 3. Accumulation of carbon over 14.6-31.10 period. Positive values indicate C loss from the ecosystem, and vice versa. Gain in delaying harvest date is lost later in the examined period.

While this is a toy demonstration for discussion here, it exemplifies types of experiments that are being designed by our team at the moment. Here we used a static and arbitrary one week delay and considered only yield, LAI and NEE variables, but more dynamic, systematic and longer-term explorations are also being devised. In addition, other constraints, such as the practical flexibility in the agricultural activities of an average farm should also be taken into account. We are planning to utilize the shiny app we developed for compiling information from farmers regarding their field activities and their flexibilities. Until such information is ready, we would respectfully prefer to omit this case study from the main text of the manuscript as not to mislead any readers, but we could provide it in an appendix or supplementary material and provide more discussion in the main text regarding the usefulness of 15-days forecasts as elaborated here. Overall, we deem that undertaking such experiments would not be possible without establishing this operative iterative forecasting system in the first place whose design and description was our main scope in this manuscript. We will clarify the introduction and the main scope of the manuscript in the revised version.

More problematic, I consider that the approach suffers from critical flaws concerning both the modeling approach and the in-situ monitoring. Therefore, I do not recommend this article for publication.

Specific comments:

My main concern is that the modelling approach doesn't seem to be mature and it has not proved its ability to simulate the carbon budget components (biomass, biomass restitution to the soil, $CO_2$ fluxes, carbon budget) in the context of this observatory. More

generally, I have strong doubts concerning the scalability of this modelling approach. First, the modelling approach should be described more in details. For instance, are simulations done at 10m resolution by assimilating LAI time series to account for spatial variability in vegetation development which can be quite significant even within a field? How is remote sensing used to calibrate the model exactly?  Which parameters are calibrated?  Is the calibration procedure parcimonious?

We thank the Reviewer for this comment. The scope of our manuscript is more focused on the description of data flows and presentation of the operational 15-day C-flux forecast system. Hence, we didn't provide much details on calibration but cited the Fer et al. (2018) paper that contains a detailed description of the calibration protocol which we also followed in this study. We only briefly mentioned that we calibrated the model because state data assimilation (forecast-analysis) techniques are best applied with calibrated model parameters (also see Huang et al., 2021). Per Reviewer's request we will add more details in the main text and provide further material in the appendix.

The BASGRA model is not spatially explicit and in this case it was run at the field-scale to represent the eddy covariance (EC) field. The LAI time series were prepared to represent the whole block accordingly by accounting the spatial variability in pixels in the observation uncertainty (P11 L18 and also discussed later in this document related to reviewers comment regarding the observational uncertainty). The parameters of the model were screened with a global sensitivity analysis by varying the parameter values in their prior range. Priors are probability distributions encapsulating the likely values of model parameters that are formulated according to information coming from previous analyses, the literature, meta analyses, expert opinion, ecological theory etc. In this study priors are chosen according to values provided by model developers and our expert opinion (also see Höglind et al., 2016; Hjelkrem et al., 2017, Huang et al., 2021). The top 20 parameters (see Table 1 below) that contribute most to model NEE and LAI output uncertainties were chosen for targeting in the calibration. Both the modeling and calibration approaches as implemented in PEcAn software are scalable and state-of-the-art techniques (Fer et al., 2018; 2021). Calibration is done by assimilating both the LAI and NEE data streams simultaneously. The influence of NEE time-series were corrected for effective sample size as described in Fer et al. (2018). Due to the coarser time step and smaller sample size of the LAI data no additional autocorrelations were applied for this data stream.

Table 1. BASGRA Parameters targeted in the calibration.

| BASGRA parameter | Description | Unit | Prior |
|---|---|---|---|
| RUBISC | Rubisco content of upper leaves | g m-2 leaf | Unif[3,12] |
| LAICR | LAI above which shading induces leaf senescence | m2 leaf m-2 | Unif[2, 7] |
| Dparam | Constant in the calculation of dehardening rate | °C-1 d-1 | Unif[0.0005, 0.0018] |
| KRESPHARD | Carbohydrate requirement of hardening | g C g-1 C °C-1 | Lnorm[-3.39, 0.22] |

| NCSHMAX | Maximum N-C ratio of shoot | g N g-1 C | Unif[0.015, 0.06] |
|---|---|---|---|
| TCSOMF | Time constant of fast SOM decomposition at 10℃ | d | Gamma[130, 0.055] |
| TMAXF | Temperature at which decomposition is maximal | ℃ | Weibull[26, 0.53] |
| TSIGMAF | Resilience of decomposition to temperature change | ℃ | Gamma[100, 4.5] |
| LAIEFT | Decrease in tillering with leaf area index | m2 leaf m-2 | Unif[0.01, 0.4] |
| NELLVM | Number of elongating leaves per non-elongating tiller | tiller-1 | Unif[1, 3] |
| RDRSCO | Relative death rate of leaves and non-elongating tillers due to shading when LAI is twice the threshold (LAICR) | d-1 | Unif[0.03, 0.13] |
| LAITIL | Maximum ratio of tiller and leaf appearance at low leaf area index | - | Unif[0.3, 1.2] |
| RDRTMIN | Minimum relative death rate of foliage | d-1 | Beta[20, 1250] |
| TCNUPT | Time constant of soil mineral N uptake | d | Unif[10, 70] |
| K | PAR extinction coefficient | m2 leaf m-2 | Unif[0.5, 0.95] |
| FLITTSOMF | Fraction of decomposed litter becoming fast SOM | g g-1 | Beta[170, 80] |
| TBASE | Minimum value of effective temperature for leaf elongation | ℃ | Unif[0.1, 6] |
| TRANCO | Transpiration coefficient | mm d-1 | Lnorm[1.5, 0.45] |
| SLAMAX | Maximum SLA of new leaves | m2 leaf kgC-1 | Norm[40, 2] |
| FSOMFSOMS | Fraction of decomposed fast SOM | g g-1 | Beta[10, 250] |

Also it is obvious that the model will perform better for simulating LAI and $CO_2$ fluxes when those variables are used for calibration! But what will be the capacity of the model to simulate not only the net $CO_2$ fluxes but also the other components of the C budget when $CO_2$ is not used for calibration (i.e. at all the ACA sites or even at larger scale)? More generally, what is the plan for applying this approach to sites not equipped with EC? What would be the accuracy of the model then to simulate the C budget components? What is the plan for validating the other C budget components (e.g. biomass) or the C budget itself when upsacling or applying the model at the ACA sites?

We agree with the Reviewer on these points and we will revise the manuscript to make our logic and plans clearer. Our main reason for performing the calibration was because data assimilation (forecast-analysis cycle) works best if the model is calibrated and initialized for the site (otherwise, when the model is too uncertain data assimilation will give almost all weight to the observations and we would learn less about the model, not to mention that the 15-day forecasts will also be more uncertain). The second reason why this was useful is that, we calibrated the model for the EC field at Qvidja but we are using it to perform simulations on the Qvidja ACA fields that are nearby as well (a work under preparation with a focus on the cutting height experiment carried out at the Qvidja ACA fields). Furthermore, we currently also have offline simulations for other Finnish mineral grassland sites that show using the parameters calibrated against Qvidja EC tower improves the model performance at these sites that are not equipped with EC. This is again part of an ongoing work with a factorial simulation design where we test the performance of the model in capturing different C budget components when informed by data streams other than CO2 fluxes (such as LAI, yield, soil moisture and soil properties that are available from all ACA sites and virtually can be made available from everywhere). We agree with the Reviewer that this is a very important question to answer (albeit not in this manuscript), hence we needed the site network, data flows and mechanics described in this paper in place to carry out these multi-site in-depth analyses.

Indeed I really doubt that soil sampling strategy will allow to validate the C budget estimates from the model at the ACA sites and it won't be feasible at the flux sites. Indeed:

- Given the soil sampling methodology described p7 I have strong doubts concerning the ability to detect SOC stock changes at the ACA sites which is the main objective of the FiON observatory. Indeed, many studies showed that between hundreds and thousands of samples are needed to detect changes in SOC stocks at a few years of intervals. See for instance the soil sampling protocols at the ICOS flux tower sites.

We appreciate the comment and we acknowledge the challenges in validating C budget estimates with soil sampling methodology. We are well aware that due to natural variation in the field, the smaller the change needed to be detected, the higher the number of replicate samples needed. We also fully agree that FiON needs regular highly intensive soil sampling campaigns for model calibration purposes.This will be included in the FiON development plans.

However, we also believe that the sampling approach that so far has been conducted serves the modelling in two crucial ways: 1) it defines the level of SOC content in the particular site (SOM content varies in ACA farms from 2% to 13%), and 2) it links in a general level soil C content to soil particle size distribution (analysis on-going) that can be used as a proxy to scale our measurements to other sites (in case no soil data is available).

We understand the limitations in our field sampling and we agree with the main criticism by the Reviewer. We will take the intensive sampling into account when planning future FiON activities.

In addition to the ACA annual monitoring, there is a pre- and post- sampling protocol for each site. The pre-samples in 2018 were taken from three georeferenced points per field with 10 cores on a 10 m radius of the center point. That gives three pooled samples per field, consisting of 10 subsamples. The samples were taken using a fixed diameter core (c.a. 15 mm), allowing estimation of soil bulk density. The sampling will be repeated in 2023 from the same points. The aim is to measure the difference in C change between test and control fields, so in total 60 subsamples are used per sampling year to make that assessment, and the subsamples are collected on fixed GPS locations, minimizing variability.

We will clarify the soil sampling methodology in the revised manuscript.

- The eddycovariance (EC) setups do not allow to quantify annual $CO_2$ fluxes nor annual C budgets at the Qvidja and Ruukki sites. Indeed, it is written p8 line 45 that the height of eddycovariance (EC) setup at the Qvidja was installed at 2.3 m height, (in Figure 1 this measurement height seems even lower). However, when a distance of 2m between the EC setup and the canopy is not respected, there is a strong risk of underestimation of the $CO_2$ Figure 1 shows that this minimum distance is clearly not respected. Also, from what is written lines 45-46, I understand that measurements are not performed over a whole cropping year at the Ruukki site (they start on June 13 2019 and end on November $4_{th}$ 2019) meaning that 1) it is not possible to evaluate the plot annual C budget and 2) it is not possible to evaluate the model's ability to simulate soil respiration outside the cropping period.

We are following and fulfilling the EC measurement guidelines provided by ICOS (*Sabbatini, S., & Papale, D., 2017*, page 7, "System height"):

"For grasslands, croplands, and shrublands with hc (canopy height) not higher than 1.75 m, hm (measurement height) must be comprised between [hm= 1.67 × hc] and [hm= 6 × hc]. For forested or more structurally complex ecosystems, hm should be between[hm= 1.67 × hc]and [hm= 2 × hc]. Anyhow, hm cannot be lower than 2 m."

For our measurement height of 2.3 m, the maximum canopy height would thus be 1.38 m ( = 2.3 m / 1.67). This is never exceeded at our study sites.

We agree that in 2019 at Ruukki, we are lacking measurements from the early growing season since the measurements started on June 13th, and hence the points 1) and 2) are not possible for the year 2019 with standard gap-filling procedures. However, in 2020 and 2021 we have full cropping year data. Also as clarification, the 2019 measurements in Ruukki did not end on November 4th; only the tower height was changed. We will clarify these in the revised manuscript.

Another concern is that the method (SEN2CORE, see P 10 line 102) for detecting clouds and could shadows for the L2A S2 data on the GEE is based on a monodate approach meaning that the performance for cloud detection is not optimal compared to methods based on multitemporal approaches (e.g. MAJA processing chain): the consequence is that NDVI and LAI time series may be quite noisy reducing the accuracy of the modelling approach relying on the use of remote sensing products times series.

We agree that using multitemporal cloud detection approaches, such as MAJA, might especially decrease the number of false negative cases (i.e. undetected cloudy images), but it is not guaranteed to be a more optimal solution. For example MAJA, to the authors' knowledge, does cloud screening at maximum with 120 m resolution. While it would probably decrease the number of false negative cases, it might simultaneously increase the number of false positive cases. The current method using the scene classification band created by SEN2COR has not produced too many outliers at our sites (some sites do not have any outliers, some might have max. 1-2 outliers per year). Above all, we do not consider the current time series noisy at any site. In addition, we assimilate LAI observations as state variables with uncertainties using Ensemble Kalman Filtering to process-based models wherefore single outliers, with optimally larger uncertainties, won't drive the model too drastically to wrong predictions. At the moment, the fact that level 2A data processed with SEN2COR is readily globally available in cloud computing platforms, such as Google Earth Engine, has also large computational benefits. But in the future we will follow and evaluate the performance of our current methodology and improve it if necessary. We will add discussion regarding our current satellite data processing chain and its caveats to the revised manuscript.

Also, why considering NDVI which is known for saturating when the vegetation is well developed (meaning that the vegetation development may be underestimated)?

We are aware and agree that NDVI saturates with well developed vegetation. NDVI is included in Field Observatory mainly for the farmers to whom NDVI is a more familiar measure compared to LAI. NDVI and its cumulative sum was requested by our farmer collaborator.

In scientific modelling work, we don't use NDVI observations at all. Instead we use LAI. As farmer collaboration is one of the main goals of FiON we intend to keep NDVI observed and visualized in the Field Observatory. We will clarify the purpose of both NDVI and LAI in the Field Observatory.

Then the following points need clarifications:

- P7 lines 10-12: it is written "At ACA sites, the measurements are done at three, static measurement points per field. The points have c.a. 30-100 m distance from each other and are located on a transect line. They were located to cover the variability of the field and cover similar soil conditions in both the test and control plots". Which variability are the authors referring to? Soil properties I assume but which soil properties where considered? SOC content ? Depth ? Texture…? Also what does "similar soil conditions" means exactly? A quantitative analysis should be provided.

The three sampling locations were chosen before any soil analysis was conducted. Therefore, the selection followed an idea of random sampling. In case there were some potential sources of variation (slope, distance to ditches etc), they were taken into account similarly in both control and C farming plots. All samples were taken similarly, from the same depths and the field had a similar texture based on farmer's earlier soil analysis, aerial photographs and knowledge on their soils.

We apologize for the unclear wording 'Similar soil conditions'. In our case it means that as random soil sampling was conducted in two fields that were of similar soil type, we assume that the starting point for the C farming experiment was equal both at the control and the C farming plot. The original soil sample data is available at Zenodo (Mattila, 2020). We will clarify these in the revised manuscript.

- P7 line 12: Please describe the sampling procedure (depth, method for sampling, i.e. core or other…)? An appropriate soil sampling methodology is critical for estimating SOC stock changes

From each of the three randomly selected sampling locations ten 20 cm deep soil cores (core diameter 14 mm) were taken in the circumference of a circle with radius of 10 m. Those ten samples were pooled and homogenised. The pooling was done to minimize variation within one sampling location. Three locations were sampled in the control site, and three in the C farming site (N=3x2).

We want to emphasize that this sampling strategy was not designed to verify small changes in soil C stocks in time. This strategy allows us to define the SOC levels, and link the SOC to the particle size distribution in each site. A better description of the soil sampling procedure will be provided in the revised manuscript.

- P8 section 3.2: it is not clear how those measurements will be used to monitor changes in SOC stocks (especially $O_2$ and $CO_2$ concentrations in the soil). Also please provide information on the model of the $O_2$ and $CO_2$ instruments and information on the depth of measurements?

The observatory is not strictly a SOC forecasting tool, it is a field ecological state observatory. The CO2 (Vaisala G525) and O2 (Apogee SO-120) sensors (10 cm depth) allow monitoring of soil and root respiration. They were included to provide feedback to farmers and to possibly guide modeling. For example, if the CO2 concentration increases by 10 fold after a rain event, that is an indication that microbial activity has

also increased, which should be captured by the process model. We will clarify this in the revised manuscript.

- P9 line 69: I don't understand this sentence. Does it mean that different gapfilling methods are used depending on the size of the gap in the observations?

We are unfortunately not sure which sentence the Reviewer here refers to. At P9 line 69, we have the equation for the modelled respiration. Maybe the following two replies also clarify this one. (We apologize for the unclear line numbering - we did not notice that the PDF conversion capped the first digit from the line numbers.)

- P10 line 88: where does this "Emod" comes from? No mention of a modelled NEE before

The gaps in the measured NEE are filled with the sum of modelled GPP and ER (P9 L65-74 ). The Emod stands for the gap-filled/modelled NEE. This will be clarified in the revised manuscript.

- P9 line 75: I don't understand the sentence "For gap-filling, the data are divided into blocks based on the harvest dates…". Please be more explicit.

The flux data are divided into sections that start and end at harvest dates. These sections are gap-filled independently, so that fluxes before a harvest are not used to predict fluxes after a harvest. This will be clarified in the revised manuscript.

- P10-11, Section 3.5: Why standardizing cumulative NDVI sums? Also does it really make sense to consider fixed starting/ending dates for growing seasons because of inter-annual variability and North/South gradients for the sites which means that the growing seasons are probably not synchronous. Last, why computing both LAI and NDVI? This point should be clarified.

The cumulative NDVI sums are standardized with fixed integration limits in order to have them in the same absolute units between sites. This in practice makes it possible to compare them between sites to some level. Without standardization the cumulative sums would be much influenced by the availability of the first and last observations of the growing season for a site. This is determined more by the cloud cover than the actual start and end of the growing season. There is of course inter-annual variability in the growing seasons but unfortunately we cannot rely on capturing the exact start and end dates from the satellite data due to cloudiness. We think having these fixed integration limits is better than using the actual first and last observations of the year which might be off by more than a month from the true growing season limits. We will clarify this in the revised manuscript as well.

As mentioned earlier, NDVI is included in Field Observatory mainly for the farmers to whom NDVI is a more familiar measure compared to LAI. We will clarify the reason for including both LAI and NDVI in the revised manuscript.

- P11 line 113: what is the justification for multiplying the associated uncertainties by 1.645?

The value 1.645 is the Z-score for 90% confidence interval. We multiply by it to convert the uncertainties to 90% confidence interval. This will be clarified in the revised manuscript.

- P11 line 118: what do you mean by observational uncertainty?

By observational uncertainty we mean the uncertainty associated with a particular single observation (from a specific image at a certain date). It is affected by the variability of the LAI values of the individual pixels within the field at that specific date. Observational uncertainty is thus unique for each image whereas the algorithmic uncertainty is usually the same for on field (it can vary depending on the number of valid samples/pixels). We will clarify this in the revised manuscript.

Technical corrections:

P2 line 35: the authors state that "Carbon farming practices include methods, such as reduced soil disturbance (reduced or zero tillage)…" while recent meta-analysis showed that soil work mostly act on the SOC

Based on Paustian et al. (2019), carbon farming practices can be classified based on whether they mainly increase C inputs or decrease C loss. The carbon farming practices applied in the ACA sites focus on increasing C inputs and no-till is not included as a tested practice (Mattila et al., 2022). This reflects the common understanding that reduced tillage only influences C distribution and not total stock. Unfortunately this sentence did not represent the experimental set up, but was more a general overview and will be revised.

P4 in Table 1: change "Crops planted to lengthen photosynthetically active period and to increase carbon assimilation, carbon and root inputs and to reduce leaching of carbon and nutrients." by "Crops planted to lengthen photosynthetically active period and to increase carbon assimilation, carbon inputs through aboveground and belowground biomass and to reduce leaching of carbon and nutrients. ». Also, the statement relative to soil amendment "Exogenous carbon input. In addition may stimulate plant growth throughincreased water holding capacity, nutrients, etc." is also true for cover crops or any practices allowing SOC stock increases.

Will be revised as suggested. However, the SOC stock increase from cover crops (300 kg C/ha/a, Poeplau and Don, 2015) is minor compared to the inputs from soil amendments tested in ACA (c.a. 10 000 kg C/ha). Therefore the soil amendments are likely the only carbon management action, which can increase water holding capacity or nutrient concentrations in 3-5 years.

P4 line 92: replace "fluxes and weather(see Sect. 3)." by "fluxes and weather (see Sect. 3)."

Thank you! We will correct this in the revised manuscript.

**References**

Corrado, S., Castellani, V., Zampori, L., and Sala, S.: Systematic analysis of secondary life cycle inventories when modelling agricultural production: A case study for arable crops, Journal of Cleaner Production, 172, 3990–4000, https://doi.org/10.1016/j.jclepro.2017.03.179, 2018.

Dietze, M. C., Fox, A., Beck-Johnson, L. M., Betancourt, J. L., Hooten, M. B., Jarnevich, C. S., Keitt, T. H., Kenney, M. A., Laney, C. M., Larsen, L. G., Loescher, H. W., Lunch, C. K., Pijanowski, B. C., Randerson, J. T., Read, E. K., Tredennick, A. T., Vargas, R., Weathers, K. C., and White, E. P.: Iterative near-term ecological forecasting: Needs, opportunities, and challenges, Proc Natl Acad Sci USA, 115, 1424–1432, https://doi.org/10.1073/pnas.1710231115, 2018.

Fer, I., Kelly, R., Moorcroft, P. R., Richardson, A. D., Cowdery, E. M., and Dietze, M. C.: Linking big models to big data: efficient ecosystem model calibration through Bayesian model emulation, Biogeosciences, 15, 5801–5830, https://doi.org/10.5194/bg-15-5801-2018, 2018.

Fer, I., Gardella, A. K., Shiklomanov, A. N., Campbell, E. E., Cowdery, E. M., De Kauwe, M. G., Desai, A., Duveneck, M. J., Fisher, J. B., Haynes, K. D., Hoffman, F. M., Johnston, M. R., Kooper, R., LeBauer, D. S., Mantooth, J., Parton, W. J., Poulter, B., Quaife, T., Raiho, A., Schaefer, K., Serbin, S. P., Simkins, J., Wilcox, K. R., Viskari, T., and Dietze, M. C.: Beyond ecosystem modeling: A roadmap to community cyberinfrastructure for ecological data-model integration, Glob. Change Biol., 27, 13–26, https://doi.org/10.1111/gcb.15409, 2021.

Hjelkrem, A.-G. R., Höglind, M., van Oijen, M., Schellberg, J., Gaiser, T., and Ewert, F.: Sensitivity analysis and Bayesian calibration for testing robustness of the BASGRA model in different environments, Ecological Modelling, 359, 80–91, https://doi.org/10.1016/j.ecolmodel.2017.05.015, 2017.

Höglind, M., Van Oijen, M., Cameron, D., and Persson, T.: Process-based simulation of growth and overwintering of grassland using the BASGRA model, Ecological Modelling, 335, 1–15, https://doi.org/10.1016/j.ecolmodel.2016.04.024, 2016.

Huang, X., Zhao, G., Zorn, C., Tao, F., Ni, S., Zhang, W., Tu, T., and Höglind, M.: Grass modelling in data-limited areas by incorporating MODIS data products, Field Crops Research, 271, 108250, https://doi.org/10.1016/j.fcr.2021.108250, 2021.

Mattila, T.: Carbon action MULTA Finnish carbon sequestration experimental field dataset 2019, https://doi.org/10.5281/ZENODO.3670654, 2020.

Mattila, T. J., Hagelberg, E., Söderlund, S., and Joona, J.: How farmers approach soil carbon sequestration? Lessons learned from 105 carbon-farming plans, Soil and Tillage Research, 215, 105204, https://doi.org/10.1016/j.still.2021.105204, 2022.

Pastorello, G., Trotta, C., Canfora, E., Chu, H., Christianson, D., Cheah, Y.-W., Poindexter, C., Chen, J., Elbashandy, A., Humphrey, M., Isaac, P., Polidori, D., Reichstein, M., Ribeca, A., van Ingen, C., Vuichard, N., Zhang, L., Amiro, B., Ammann, C., Arain, M. A., Ardö, J., Arkebauer, T., Arndt, S. K., Arriga, N., Aubinet, M., Aurela, M., Baldocchi, D., Barr, A., Beamesderfer, E., Marchesini, L. B., Bergeron, O., Beringer, J., Bernhofer, C., Berveiller, D., Billesbach, D., Black, T. A., Blanken, P. D., Bohrer, G., Boike, J., Bolstad, P. V., Bonal, D., Bonnefond, J.-M., Bowling, D. R., Bracho, R., Brodeur, J., Brümmer, C., Buchmann, N., Burban, B., Burns, S. P., Buysse, P., Cale, P., Cavagna, M., Cellier, P., Chen, S., Chini, I., Christensen, T. R., Cleverly, J., Collalti, A., Consalvo, C., Cook, B. D., Cook, D., Coursolle, C., Cremonese, E., Curtis, P. S., D'Andrea, E., da Rocha, H., Dai, X., Davis, K. J., Cinti, B. D., Grandcourt, A. de, Ligne, A. D., De Oliveira, R. C., Delpierre, N., Desai, A. R., Di Bella, C. M., Tommasi, P. di, Dolman, H., Domingo, F., Dong, G., Dore, S., Duce, P., Dufrêne, E., Dunn, A., Dušek, J., Eamus, D., Eichelmann, U., ElKhidir, H. A. M., Eugster, W., Ewenz, C. M., Ewers, B., Famulari, D., Fares, S., Feigenwinter, I., Feitz, A., Fensholt, R., Filippa, G., Fischer, M., Frank, J., Galvagno, M., et al.: The FLUXNET2015 dataset and the ONEFlux processing pipeline for eddy covariance data, Scientific Data, 7, 225, https://doi.org/10.1038/s41597-020-0534-3, 2020.

Paustian, K., Larson, E., Kent, J., Marx, E., and Swan, A.: Soil C Sequestration as a Biological Negative Emission Strategy, Front. Clim., 1, 8, https://doi.org/10.3389/fclim.2019.00008, 2019.

Petchey, O. L., Pontarp, M., Massie, T. M., Kéfi, S., Ozgul, A., Weilenmann, M., Palamara, G. M., Altermatt, F., Matthews, B., Levine, J. M., Childs, D. Z., McGill, B. J., Schaepman, M. E., Schmid, B., Spaak, P., Beckerman, A. P., Pennekamp, F., and Pearse, I. S.: The ecological forecast horizon, and examples of its uses and determinants, Ecol Lett, 18, 597–611, https://doi.org/10.1111/ele.12443, 2015.

Poeplau, C. and Don, A.: Carbon sequestration in agricultural soils via cultivation of cover crops – A meta-analysis, Agriculture, Ecosystems & Environment, 200, 33–41, https://doi.org/10.1016/j.agee.2014.10.024, 2015.

Sabbatini, S. and Papale, D.: ICOS Ecosystem Instructions for Turbulent Flux Measurements of CO2, Energy and Momentum, https://doi.org/10.18160/QWV4-639G, 2017.

Scharf, P. C., Shannon, D. K., Palm, H. L., Sudduth, K. A., Drummond, S. T., Kitchen, N. R., Mueller, L. J., Hubbard, V. C., and Oliveira, L. F.: Sensor‑Based Nitrogen Applications Out‑Performed Producer‑Chosen Rates for Corn in On‑Farm Demonstrations, Agron.j., 103, 1683–1691, https://doi.org/10.2134/agronj2011.0164, 2011.

Sela, S., Woodbury, P. B., and van Es, H. M.: Dynamic model-based N management reduces surplus nitrogen and improves the environmental performance of corn production, Environ. Res. Lett., 13, 054010, https://doi.org/10.1088/1748-9326/aab908, 2018.

Viskari, T., Laine, M., Kulmala, L., Mäkelä, J., Fer, I., and Liski, J.: Improving Yasso15 soil carbon model estimates with ensemble adjustment Kalman filter state data assimilation, Geosci. Model Dev., 13, 5959–5971, https://doi.org/10.5194/gmd-13-5959-2020, 2020.